# ImpatientCapsAndRuns: Approximately Optimal Algorithm Configuration from an Infinite Pool

**Gellért Weisz**
Deepmind/University College London, UK
gellert@google.com

**András György**
Deepmind, UK
agyorgy@google.com

**Wei-I Lin**
University of British Columbia, Canada
weiilin@students.cs.ubc.ca

**Devon Graham**
University of British Columbia, Canada
drgraham@cs.ubc.ca

**Kevin Leyton-Brown**
University of British Columbia, Canada
kevinlb@cs.ubc.ca

**Csaba Szepesvári**
Deepmind/University of Alberta, Canada
szepi@google.com

**Brendan Lucier**
Microsoft Research, USA
brlucier@microsoft.com

## Abstract

Algorithm configuration procedures optimize parameters of a given algorithm to perform well over a distribution of inputs. Recent theoretical work focused on the case of selecting between a small number of alternatives. In practice, parameter spaces are often very large or infinite, and so successful heuristic procedures discard parameters "impatiently", based on very few observations. Inspired by this idea, we introduce IMPATIENTCAPSANDRUNS, which quickly discards less promising configurations, significantly speeding up the search procedure compared to previous algorithms with theoretical guarantees, while still achieving optimal runtime up to logarithmic factors under mild assumptions. Experimental results demonstrate a practical improvement.

## 1 Introduction

Solvers for computationally hard problems (e.g., SAT, MIP) often expose many parameters that only affect runtime rather than solution quality. Choosing values for these parameters is seldom easy or intuitive, and different settings can lead to drastically different runtimes—days versus seconds—for a given input instance. Such parameters are exposed in the first place because they do not have known, globally optimal settings, instead typically expressing tradeoffs between different heuristic mechanisms or implicit assumptions about problem structure. In practice, solver end-users typically need to repeatedly solve similar problems: e.g., integer programs modeling airline crew scheduling problems; or SAT formulae used to formally verify a sequence of related hardware or software designs. This gives rise to the problem of *algorithm configuration*: finding a joint setting of parameters for a given algorithm so that it performs well on input instances drawn from a given distribution. We make no restrictions on the space of possible parameters or its structure: they may be continuous, categorical, subject to arbitrary constraints, and may contain jump discontinuities. We refer to a joint setting of all the algorithm's parameters as a *configuration* to stress this generality. A common

metric of performance for a configuration, and the one we consider in this work, is mean runtime: we prefer configurations that are faster, on average, on the problems we care about solving. An algorithm configuration method can sample instances from the distribution underlying an application and can run any configuration (possibly also sampled from the set possible configurations) on any sampled instance until a timeout of its choice, and the goal is to find a configuration with nearly optimal mean runtime while using the least amount of time during the search.[1]

Heuristic methods for algorithm configuration such as ParamILS [18, 19], GGA [2, 3], irace [11, 29] and SMAC [21, 22] have been used with great success for more than a decade, but they do not come with any rigorous performance guarantees. More recently, algorithm configuration has also been considered from a theoretical perspective. Kleinberg et al. [24] introduced a framework to analyze algorithm configuration methods theoretically, and presented the first configuration procedure, STRUCTURED PROCRASTINATION (SP), which is guaranteed to find an approximately optimal solution with a non-trivial worst-case runtime bound. Since then algorithms with better theoretical guarantees have been developed [35, 36, 25]. Overall, these theoretically-motivated configuration procedures have nice properties, such as achieving near-optimal asymptotic worst-case running times. However, none of them yet achieves competitive performance on practical problem benchmarks, for two key reasons: (i) heuristic methods usually iteratively select candidate configurations that appear likely to perform well given previous samples from the configuration space (e.g., leveraging structure in the parameter space, such as smoothness or low pseudodimension [20, 30]), whereas the theoretical algorithms select configurations randomly; and (ii) heuristic methods often impatiently discard less promising configurations based on just a few runtime observations, while the theoretical algorithms are more conservative and continue evaluating them until they demonstrate, with high probability, that another configuration is better. Such early discard strategies are particularly effective when the configuration space contains one or a few configurations that drastically outperform all others. This "needle-in-a-haystack" scenario is common in practice, perhaps in part explaining the success of these heuristic methods.

In this paper we take a significant step towards theoretically grounded *and* practical algorithm configuration by addressing the second problem. We build on CAPSANDRUNS (CAR) [36], a simple and intuitive algorithm that continuously discards configurations that perform poorly relative to a global upper bound on the best achievable mean runtime. Here we introduce IMPATIENTCAPSAN-DRUNS (ICAR), which equips CAR with the ability to quickly discard less-promising configurations by applying an initial "precheck" mechanism that allows poorly performing configurations to be discarded quickly. Additionally, via a more careful analysis we are able speed up a key subroutine from CAR. While ICAR retains the favorable optimality and runtime guarantees of CAR under mild assumptions, it is also provably faster in needle-in-a-haystack scenarios where most configurations are considerably weaker than the best ones (these are the cases where good algorithm configuration procedures are the most useful, because identifying a good configuration is the most consequential.) Because of its precheck procedure, ICAR is able to examine more configurations than CAR, and hence finds configurations with better mean runtime. Furthermore, not wasting time on examining bad configurations, the total runtime of ICAR is significantly smaller than that of CAR and any other existing procedure with theoretical guarantees, making a step towards closing the performance gap relative to heuristic procedures.

Finally, we briefly survey some less closely related work. Gupta & Roughgarden [14] initiated the study of algorithm configuration from a learning-theoretic perspective. Rather than seek general purpose configuration procedures, as we do in this work, this and subsequent approaches seek to bound the number of training samples required to guarantee good generalization for specific classes of problems. Examples include combinatorial partitioning problems such as max-cut and clustering [6], branching strategies in tree search algorithms [7], and general algorithm configuration when the runtime is piecewise-constant over its parameter space [8]. Hyperparameter-search methods based on multi-armed bandit algorithms are also related. The main difference is that this literature focuses on settings where every configuration run costs the same amount or where there is a tradeoff between how long each configuration is run and the accuracy with which its performance is estimated [5, 28]; thus, these methods do not face questions like how many instances to consider and how to cap runs.

The rest of the paper is organized as follows. The formal model of algorithm configuration is given in Section 2. The ICAR algorithm is presented and analyzed in Section 3. Experiments on some algorithm configuration benchmarks are given in Section 4. Proofs and additional experimental results are deferred to the appendix.

## 2   The Model

Following Kleinberg et al. [24], the algorithm configuration problem is defined by a triplet $(\Pi, \Gamma, R)$, where $\Pi$ is a distribution over possible configurations, $\Gamma$ is a distribution over input instances, and $R(i, j)$ is the runtime of a configuration $i$ on a problem instance $j$. For example $\Pi$ and $\Gamma$ may simply be uniform distributions, respectively over the space of hyperparameters and the set of past problem instances seen. The mean runtime of a configuration $i$ is defined as $R(i) = \mathbb{E}_{j \sim \Gamma}[R(i, j)]$, and the ultimate goal of an algorithm configuration method is to find a configuration $i$ minimizing $R(i)$.

During this search the configuration method needs to explore new configurations, which can be sampled from $\Pi$.[2] The configuration method can also sample problem instances from $\Gamma$ and run a configuration $i$ on an instance $j$ until it finishes, or the execution time exceeds a specified timeout $\tau \geq 0$. The use of such a timeout allows for a tradeoff between learning more about the runtime of a single configuration–instance pair and considering a larger number of such pairs.

To this end, for any configuration $i$ we consider the $\tau$-capped expected runtime $R_\tau(i) = \mathbb{E}_{j \sim \Gamma}[\min\{R(i, j), \tau\}]$. Furthermore, for any $\delta \in (0, 1)$, let $t_\delta(i) = \inf_t\{t \,:\, \Pr_{j \sim \Gamma}(R(i, j) > t) \leq \delta\}$ denote the $\delta$-quantile of $i$'s runtime, and define $R^\delta(i) = R_{t_\delta(i)}(i)$ the $\delta$-capped expected runtime of $i$.[3] That is, $R^\delta(i)$ is the mean runtime of $i$ if we cap the slowest $\delta$-fraction of its runtimes.

Since a globally optimal configuration may be arbitrarily hard to find, we instead seek a solution that is competitive with the performance of the top $\gamma$-fraction of the configurations for a $\gamma \in (0, 1)$. That is, instead of finding a configuration close to $\mathrm{OPT} = \min_i\{R(i)\}$, we search for one close to $\mathrm{OPT}^\gamma = \inf_{x \in \mathbb{R}^+}\{x \,:\, \Pr_{i \sim \Pi}(R(i) \leq x) \geq \gamma\}$. Additionally, since the average runtime of any configuration, including the optimal one, could be totally dominated by a few incredibly unlikely but arbitrarily large runtime values, we seek solutions whose expected $\delta$-capped runtime is close to the $\delta$-capped optimum. However, it turns out that this relaxed property is still impossible to verify [35]. Following Weisz et al. [35], we address this by adding a small amount of slack to the benchmark, comparing to the $(\delta/2)$-capped optimum rather than the $\delta$-capped optimum. Putting this together, we seek solutions whose expected $\delta$-capped runtime is close to the $(\delta/2)$-capped optimum, after excluding the best $\gamma$-fraction of configurations: $\mathrm{OPT}^\gamma_{\delta/2} = \inf_{x \in \mathbb{R}^+}\left\{x \,:\, \Pr_{i \sim \Pi}[R^{\frac{\delta}{2}}(i) \leq x] \geq \gamma\right\}$.

**Definition 1** $((\varepsilon, \delta, \gamma)$-optimality). *A configuration $i$ is $(\varepsilon, \delta, \gamma)$-optimal if $R^\delta(i) \leq (1 + \varepsilon)\mathrm{OPT}^\gamma_{\delta/2}$.*

This definition generalizes the notion of $(\varepsilon, \delta)$-optimality of Weisz et al. [36] for a finite set of configurations, where instead of the top-$\gamma$ portion, we aim to achieve the performance of the best configuration (up to $\varepsilon$): for a finite set of $N$ configurations, configuration $i$ is $(\varepsilon, \delta)$-optimal if it is $(\varepsilon, \delta, 1/N)$-optimal when $\Pi$ is the uniform distribution over the $N$ configurations.

## 3   The Algorithm

Recent theoretically-sound algorithm configuration procedures make several runtime measurements for every configuration in a finite pool $\mathcal{N}$, and stop when they can confirm, with high probability, that one configuration is close enough to the best one. The main challenge is to avoid wasting time on (a) hard input instances with large runtimes; and (b) bad configurations that will be eliminated later. To this end, STRUCTURED PROCRASTINATION (SP) [24] and its improved version STRUCTURED PROCRASTINATION WITH CONFIDENCE (SPC) [25] gradually increase the runtime cap for every configuration-instance pair, while carefully determining an order to evaluate these pairs, depending on the configurations' empirical average runtime (SP) or empirical confidence bounds on the mean runtimes (SPC). LEAPSANDBOUNDS (LAB) [35], which introduced empirical confidence bounds to

the algorithm configuration problem, works with a much simpler schedule, and tests all configurations for a given time budget, which is increased gradually.

On the other hand, CAPSANDRUNS (CAR) [36] first measures the runtime cap for each configuration guaranteeing that at least a $(1 - \delta)$-portion of the instances can be solved within that cap, then runs a racing algorithm (based on continuously recomputing confidence bounds on the mean runtimes) to select which capped configuration is the best. During the race, all configurations are run in parallel on more and more problem instances, and their mean runtime is continuously estimated. This makes it possible to maintain a high-probability upper bound $T$ on the optimal capped runtime, and any configuration with a runtime lower bound above $T$ can be eliminated. The algorithm stops when it can prove that a configuration is $(\varepsilon, \delta)$-optimal.

To apply any of the above methods to an infinite pool of configurations, one can simply select a pool of $\left\lceil \frac{\log(\zeta)}{\log(1-\gamma)} \right\rceil$ configurations randomly from $\Pi$ to ensure that with probability at least $1 - \zeta$ it contains a configuration that belongs to the top $\gamma$-fraction of all the configurations. Thus the above methods can select $(\varepsilon, \delta, \gamma)$-optimal configurations from an infinite pool, with attractive theoretical guarantees. Our focus in this paper is on extending CAR, due to its conceptual simplicity and good practical performance. However, in contrast to LAB and SPC, which try to assign little runtime to bad configurations from the very beginning, at the start CAR spends the same amount of time testing all configurations. This is because the estimation of the runtime caps is done in parallel, so every configuration is run for an equally long time until the first cap is found for any configuration (only after this can the algorithm start eliminating configurations with large mean runtimes). As a result, CAR spends more time testing the worst configurations than LAB or SPC. Appendix B further compares these methods and their runtime bounds.

IMPATIENTCAPSANDRUNS (ICAR) addresses this problem, introducing a "precheck" mechanism to ensure that bad configurations are eliminated early. The PRECHECK function estimates the mean capped runtime (up to a constant multiplicative factor) needed by a configuration to solve at least a constant fraction of the problem instances (less than $1 - \delta/2$). If this capped runtime is large compared to the upper bound $T$ on the $(\varepsilon, \delta, \gamma)$-optimal runtime (maintained similarly as in CAR), the configuration is rejected and eliminated from further analysis. This procedure is very similar to the CAR algorithm (with some fixed, constant $\varepsilon$ and $\delta$); only the specific rejection conditions differ mildly. Note that the runtime estimated by PRECHECK is a lower bound to the $\delta/2$-capped runtime, ensuring that good configurations are unlikely to be rejected. The efficiency of PRECHECK crucially depends on the quality of the bound $T$ on the optimal runtime. Therefore, similarly to SPC, ICAR gradually introduces more and more configurations in batches $\mathcal{N}_k, k = K - 1, \ldots, 0$: if a configuration passes PRECHECK, a (rough) estimate of its capped runtime is calculated (up to a multiplicative constant, for a cap slightly larger than the $\delta$ quantile), again by first measuring the runtime cap, then estimating the mean runtime using the measured cap. This runtime estimate is then used to reduce the bound $T$, which improves the performance of PRECHECK for the next batch of configurations, $\mathcal{N}_{k-1}$. The size of batch $\mathcal{N}_k$ is of order $1/\gamma_k$ with $\gamma_k = 2^k \gamma$, ensuring that with high probability it contains an $(\varepsilon, \delta, \gamma_k)$-optimal configuration (whose mean runtime is then bounded by $\text{OPT}_{\delta/2}^{\gamma_k}$). As a consequence, after batch $\mathcal{N}_k$, $T$ is at most $2\text{OPT}_{\delta/2}^{\gamma_k}$, gradually reducing towards $2\text{OPT}_{\delta/2}^{\gamma}$. Finally, the racing part of CAR is run over all surviving configurations, further reducing $T$ towards $\text{OPT}_{\delta/2}^{\gamma}$, and stopping when an $(\varepsilon, \delta, \gamma)$-optimal configuration is found.

Now we are ready to present the main theoretical result of the paper, a performance guarantee for ICAR. The components of the algorithm are presented in Algorithms 1–5. We then discuss each and present a proof sketch for the theorem (the detailed proof is given in Appendix A).

**Theorem 1.** *For input parameters $\varepsilon \in (0, 1/3), \delta \in (0, 0.2), \gamma \in (0, 1)$, integer $K \geq 1$, and failure parameter $\zeta \in (0, 1/12)$, with probability at least $1 - 12\zeta$, IMPATIENTCAPSANDRUNS finds an $(\varepsilon, \delta, \gamma)$-optimal configuration with total work[4] bounded by[5]*

$$\tilde{\mathcal{O}}\left( \frac{\text{OPT}_{\delta/2}^{\gamma}}{\varepsilon^2 \delta \gamma} \cdot F(38\text{OPT}_{\delta/2}^{\gamma}) + \sum_{k=0}^{K-2} \frac{\text{OPT}_{\delta/2}^{\gamma_k}}{\gamma_k} \left( 1 + \frac{F(38\text{OPT}_{\delta/2}^{\gamma_{k+1}})}{\delta} \right) + \frac{\text{OPT}_{\delta/2}^{\gamma_{K-1}}}{\delta\gamma_{K-1}} \right), \quad (1)$$

*where $\gamma_k = 2^k \gamma$, and $F(x) = \Pr_{i \sim \Pi}(R^{0.35}(i) \leq x) + 4\zeta/K$.*

**Global variables**

1: Instance distribution $\Gamma$
2: Phase I measurements count $b$
3: $T \leftarrow \infty$  ▷ Upper bound on $\mathrm{OPT}^{\gamma}_{\delta/2}$, updated continuously by all parallel processes
4: Set $\overline{\mathcal{N}}$ of algorithm configurations

---

**Algorithm 1** IMPATIENTCAPSANDRUNS

1: **Inputs:** Precision parameter $\varepsilon \in (0, \frac{1}{3})$, Quantile parameter $\delta \in (0, \frac{1}{7})$, Optimality quantile target parameter $\gamma$, Failure probability parameter $\zeta \in (0, \frac{1}{12})$, Number of iterations $K$, Instance distribution $\Gamma$, Configuration distribution $\Pi$
2: $\mathcal{N}_k \leftarrow$ Sample $\left\lceil \frac{\log(\zeta/K)}{\log(1-\gamma_k)} \right\rceil - \left\lceil \frac{\log(\zeta/K)}{\log(1-\gamma_{k+1})} \right\rceil$ many configurations from $\Pi$ for $k \in [0, K-1]$
3: $b \leftarrow \left\lceil \frac{26}{\delta} \log\left(\frac{2n}{\zeta}\right) \right\rceil$
4: Reset $T \leftarrow \infty$
5: $\mathcal{N} \leftarrow \bigcup_{k=0}^{K-1} \mathcal{N}_k$
6: **for** $k = K - 1$ downto $0$ **do**
7: $\quad \overline{\mathcal{N}}_k \leftarrow$ PRECHECK $(\mathcal{N}_k, \zeta/K)$
8: $\quad$ **for** configurations $i \in \overline{\mathcal{N}}_k$ in parallel[a] **do**
9: $\quad\quad P_i \leftarrow$ CAPSANDRUNS $(i, \varepsilon, \delta, \zeta)$ thread
10: $\quad\quad$ Start running $P_i$
11: $\quad\quad$ Pause $P_i$ when $b$ runs of RUNTIMEEST finished
12: $\quad$ **end for**
13: **end for**
14: $\overline{\mathcal{N}} \leftarrow$ PRECHECK $(\mathcal{N}, \zeta/K)$
15: Continue runing $P_i$ for $i \in \overline{\mathcal{N}}$
16: // CAPSANDRUNS eliminates the threads
17: Wait until all threads finish, abort if $|\overline{\mathcal{N}}| = 1$
18: **return** $i^* = \mathrm{argmin}_{i \in \overline{\mathcal{N}}} \bar{Y}(i)$ and $\tau_{i^*}$

---

**Algorithm 2** CAPSANDRUNS thread

1: **Inputs:** Configuration $i$, precision $\varepsilon$, quantile parameter $\delta$, failure probability parameter $\zeta$
2: // Phase I:
3: Run $\tau_i \leftarrow$ QUANTILEEST $(i, \delta)$
4: // Phase II:
5: **if** QUANTILEEST $(i, \delta)$ aborted **then**
6: $\quad$ Remove $i$ from $\overline{\mathcal{N}}$
7: **else**
8: $\quad \bar{Y}(i) \leftarrow$ RUNTIMEEST$(i, \tau_i, \varepsilon, \delta, \zeta)$
9: $\quad$ **if** RUNTIMEEST rejected $i$ **then**
10: $\quad\quad$ Remove $i$ from $\overline{\mathcal{N}}$
11: $\quad$ **end if**
12: **end if**

---

**Algorithm 3** QUANTILEEST

1: **Inputs:** $i, \delta$
2: **Initialize:** $m \leftarrow \left\lceil (1 - \frac{3}{4}\delta)b \right\rceil$
3: Run configuration $i$ on $b$ instances, in parallel, until $m$ of these complete. Abort and return *abort* if total work $\geq 1.5Tb$.
4: $\tau \leftarrow$ runtime of $m^{th}$ completed instance
5: **return** $\tau$

---

[a]When running CAPSANDRUNS threads in parallel, we allocate the same amount of time for every running thread, regardless of the number of parallel tasks they themselves may be performing.

---

**Algorithm 4** PRECHECK

1: **Inputs:** Configurations $\mathcal{M}$, error parameter $\zeta/K$
2: $\mathcal{M}' \leftarrow \{\}$  ▷ empty set
3: $b' \leftarrow \left\lceil 32.1 \log\left(\frac{2K}{\zeta}\right) \right\rceil$
4: **if** $T = \infty$ **then**
5: $\quad$ **return** $\mathcal{M}$
6: **end if**
7: **for** $i \in \mathcal{M}$ **do**
8: $\quad$ **if** $T$ last set when evaluating $i$ **then**
9: $\quad\quad$ append $i$ to $\mathcal{M}'$  ▷ Add automatically
10: $\quad\quad$ Continue
11: $\quad$ **end if**
12: $\quad$ // Phase I:
13: $\quad$ Run $i$ on $b'$ instances in parallel until $\lceil 0.8b' \rceil$ complete. Abort if total work $\geq 1.9Tb'$.
14: $\quad$ **if** not aborted **then**
15: $\quad\quad \tau' \leftarrow$ runtime of $\lceil 0.8b' \rceil^{th}$ completed instance
16: $\quad\quad$ // Phase II:
17: $\quad\quad$ **for** $l = 1, l \leq b'$ **do**
18: $\quad\quad\quad Y_l \leftarrow$ runtime of configuration $i$ on instance $j \sim \Gamma$, with timeout $\tau'$
19: $\quad\quad\quad$ **if** $\sum_{m=1}^{l} Y_m > 2.99Tb'$ **then**
20: $\quad\quad\quad\quad$ // Stop measuring if total work too large
21: $\quad\quad\quad\quad$ Break
22: $\quad\quad\quad$ **end if**
23: $\quad\quad$ **end for**
24: $\quad\quad$ Sample mean $\bar{Y} \leftarrow \frac{1}{|Y|} \sum_{y \in Y} y$
25: $\quad\quad$ Sample variance $\bar{\sigma}^2 \leftarrow \frac{1}{|Y|} \sum_{y \in Y} (y - \bar{Y})^2$
26: $\quad\quad$ Confidence $C \leftarrow \bar{\sigma} \sqrt{\frac{2 \log(\frac{3K}{\zeta})}{l}} + \frac{3\tau' \log(\frac{3K}{\zeta})}{l}$
27: $\quad\quad$ **if** $\bar{Y} - C \leq T$ **then**
28: $\quad\quad\quad$ append $i$ to $\mathcal{M}'$
29: $\quad\quad$ **end if**
30: $\quad$ **end if**
31: **end for**
32: **return** $\mathcal{M}'$

---

**Algorithm 5** RUNTIMEEST

1: **Inputs:** $i, \tau_i, \varepsilon, \delta, \zeta$
2: **Initialize:** $j \leftarrow 0$
3: **while** True **do**
4: $\quad$ Sample $j^{th}$ instance $J$ from $\Gamma$
5: $\quad Y_{i,j} \leftarrow$ runtime of configuration $i$ on instance $J$, with timeout $\tau_i$
6: $\quad$ Sample mean $\bar{Y}(i) \leftarrow \frac{1}{j} \sum_{j'=1}^{j} Y_{i,j'}$
7: $\quad$ Sample variance $\bar{\sigma}_i^2 \leftarrow \frac{1}{j} \sum_{j'=1}^{j} (Y_{i,j'} - \bar{Y}(i))^2$
8: $\quad$ // Calculate confidence:
9: $\quad C_i \leftarrow \bar{\sigma}_i \sqrt{\frac{2 \log(\frac{3nj(j+1)}{\zeta})}{j}} + \frac{3\tau_i \log(\frac{3nj(j+1)}{\zeta})}{j}$
10: $\quad$ **if** $\bar{Y}(i) - C_i > T$ **then**
11: $\quad\quad$ **return** reject $i$
12: $\quad$ **end if**
13: $\quad$ **if** j=b **then**
14: $\quad\quad T \leftarrow \min\{T, 2\bar{Y}(i)\}$.
15: $\quad$ **end if**
16: $\quad T \leftarrow \min\{T, \bar{Y}(i) + C_i\}$  ▷ upper confidence
17: $\quad$ **if** $C_i \leq \frac{\varepsilon}{3}(2\bar{Y}(i) - C_i)$ **then**
18: $\quad\quad$ **return** accept $i$ with runtime estimate $\bar{Y}(i)$.
19: $\quad$ **end if**
20: $\quad j \leftarrow j + 1$
21: **end while**

**Discussion.** *(i)* To illustrate the advantages captured by the theorem, consider a situation where configuration runtimes are distributed exponentially, with their mean distributed uniformly over an interval $[A, A + B]$. When the number of near-optimal configurations is small (i.e., $B/A$ is large enough), the bound on the fraction of configurations surviving PRECHECK, $F(38\mathrm{OPT}_{\delta/2}^{\gamma})$, roughly scales with $\gamma$, resulting in a runtime $\mathrm{OPT}_{\delta/2}^{\gamma}/(\varepsilon^2\delta)$, providing a $\gamma$-factor speedup over typical bounds in other work (which scale with $\mathrm{OPT}_{\delta/2}^{\gamma}/(\varepsilon^2\delta\gamma)$). (Details are given in Appendix C.)

*(ii)* The first term in the bound corresponds to the work done in the final racing part of ICAR. The other terms correspond to the work done for each batch $\mathcal{N}_k$ (except that the cost of the last precheck is included in the $k = 0$ term).

*(iii)* Kleinberg et al. [24] showed that to find an $(\varepsilon, \delta)$-optimal configuration out of a pool of size $n$, the worst-case minimum total runtime is $\tilde{\Omega}(\frac{n\mathrm{OPT}}{\varepsilon^2\delta})$.[6] Since we need to test $\Omega(1/\gamma)$ configurations, in the worst case the total runtime needed to find an $(\varepsilon, \delta, \gamma)$-optimal configuration is about $\frac{\mathrm{OPT}_{\delta/2}^{\gamma}}{\varepsilon^2\delta\gamma}$. The first term in our bound matches this, except that it is multiplied by (an upper bound on) the fraction of configurations surviving PRECHECK, $F(38\mathrm{OPT}_{\delta/2}^{\gamma})$. Under typical parameter settings, this is the main term of the bound—the only one scaling with $1/(\varepsilon^2\delta\gamma)$—and the performance improvement of ICAR over CAR comes from this additional factor of $F(38\mathrm{OPT}_{\delta/2}^{\gamma})$. Note that this term, and all the others, scale with a bound on the *optimal* runtime for the set of configurations they correspond to (e.g., for batch $\mathcal{N}_k$ they scale with $\mathrm{OPT}_{\delta/2}^{\gamma_k}$).

*(iv)* $F(38\mathrm{OPT}_{\delta/2}^{\gamma_{k+1}})$ is an upper bound on the number of configurations surviving PRECHECK from $\mathcal{N}_k$. Due to the a worst-case nature of our analysis, the bound is conservative, and in practice the number of surviving configurations is much smaller. In essence, this term measures how many configurations are competitive with a very good ($\mathrm{OPT}_{\delta/2}^{\gamma_{k+1}}$-optimal) configuration. In other words, it measures the "needle-in-a-haystack" property of the configuration task.

*(v)* The first term can be replaced with the problem-dependent bound of Weisz et al. [36, Equation 1] for $n = F(38\mathrm{OPT}_{\delta/2}^{\gamma})\frac{1}{\gamma}$ configurations. This bound depends on the characteristics of the runtime distributions of the configurations, and show that the algorithm can run much faster if the problem is easy, e.g., adapting to the relative variance of the runtime distributions. However, for simplicity, we only present the worst-case form here.

*(vi)* The rest of the terms represent the cost of iteratively selecting only the best configurations to evaluate. None of these terms depends on $1/\varepsilon^2$. Note $1/\gamma_k$ is roughly the number of configurations in batch $\mathcal{N}_k$, and each configuration is run essentially as long as the best configuration in that batch ($\mathrm{OPT}_{\delta/2}^{\gamma_k}$). Each of these configurations is run on constantly many instances in PRECHECK, and the surviving fraction of $F(38\mathrm{OPT}_{\delta/2}^{\gamma_{k+1}})$ configurations is also run on $1/\delta$ instances to measure an accurate cap and set the bound $T$. These terms scale with $\mathrm{OPT}_{\delta/2}^{\gamma_k}/\gamma_k = 2^{-k}\mathrm{OPT}_{\delta/2}^{\gamma_k}/\gamma$. Thus, the bound is only meaningful when $2^{-k}\mathrm{OPT}_{\delta/2}^{\gamma_k}$ is not too large. While in principle they can be infinite, in realistic scenarios this is not the case. Nevertheless, this requires the practitioner to choose $\gamma_{K-1}$ such that it guarantees a small-enough optimal runtime $\mathrm{OPT}_{\delta/2}^{\gamma_{K-1}}$, which is essentially the same task as choosing a proper $\gamma$. The terms also scale with $1/\delta$, the effect of this is mitigated by the success of PRECHECK: for $k \neq K - 1$, each term is multiplied by the upper bound $F(38\mathrm{OPT}_{\delta/2}^{\gamma_k})$ on the fraction of configurations surviving PRECHECK.

*(vii)* Our analysis shows that CAR can be sped up significantly without sacrificing any of its guarantees from Weisz et al. [36], by measuring the runtime caps on fewer samples (i.e., replacing the original value of $b$ from Weisz et al. [36] with the one in Line 3 of Algorithm 1). We call this improved algorithm CAR ++. This effect is also partly responsible for the improved performance of ICAR.

**Insights into the algorithm and proof sketch** We start with a brief description of the CAR algorithm, which runs parallel threads of Algorithm 2 for all configurations it considers. As described before, one thread, working on configuration $i$, has two phases: In the first phase, implemented in QUANTILEEST (Algorithm 3), a runtime cap $\tau_i$ is determined such that $i$ is guaranteed, with

high probability, to solve a random instance with probability between $1 - \delta$ and $1 - \delta/2$ (i.e. $t_\delta(i) \leq \tau_i < t_{\delta/2}(i)$).[7] This is achieved by solving sufficiently many instances in parallel, and $\tau_i$ is selected to be the time when a $(1 - 3\delta/4)$-fraction of the instances are solved. If measuring this cap takes too long, then QUANTILEEST stops measuring and eliminates configuration $i$. Unless this happens, in the second phase, the method RUNTIMEEST (Algorithm 3) is used to estimate the mean $\tau_i$-capped runtime $R_{\tau_i}(i)$ of $i$, by solving successively selected random instances and computing the average runtime $\bar{Y}(i)$. Then the empirical Bernstein inequality [4] is used to guarantee that $R_{\tau_i}(i) \in [\bar{Y}(i) - C_i, \bar{Y}(i) + C_i]$ for $C_i$ calculated in Line 9 of Algorithm 5. This confidence interval is used continuously in multiple ways: (i) to reduce a global upper bound $T$ on the best possible runtime of all the configurations (Line 16); (ii) to eliminate a configuration if it shows that $R_{\tau_i}(i) > T$ (Line 10); and (iii) to check if $R_{\tau_i}(i)$ is estimated accurately enough (Line 17). The procedure (which is an instance of a so-called Bernstein race [31]) continues until each configuration is either measured accurately or eliminated. The continuous elimination (also in QUANTILEEST) and parallel execution guarantees that when the procedure stops, every configuration is run for at most $\tilde{\mathcal{O}}(\text{OPT}/(\varepsilon^2\delta))$ time, and eventually an $(\varepsilon, \delta)$-optimal configuration is found, where OPT is the minimum mean $\delta/2$-quantile capped runtime of the configurations.

As explained before, ICAR (Algorithm 1) starts to examine new configurations in batches. For any batch $\mathcal{N}_k$, first each configuration is quickly tested to see if it can be excluded from the set of potentially optimal configurations. This is done by the PRECHECK function, given in Algorithm 4. PRECHECK is very similar to CAR, but works with constant accuracy and quantile parameters instead of $\varepsilon$ and $\delta$, ensuring that it runs quickly, in time independent of these parameters. Also, the conditions to reject configurations are slightly different. For a configuration $i$, PRECHECK first estimates a cap $\tau'$ that guarantees solving random instances with constant probability $p_i \in [0.1, 0.35]$; then the mean $\tau'$-capped runtime is estimated roughly up to a constant multiplicative error. Since $\delta/2 \leq 0.1$ (the lower bound on $p_i$), PRECHECK can compute multiplicative lower bounds on the runtime $R_{\delta/2}(i)$. These are then used to set the rejection conditions such that at least one of the best configurations from this batch $i$ with $R_{\delta/2}(i) \leq T$ is not rejected. Combining with the fact that $\bigcup_{j=k}^{K-1} \mathcal{N}_j$ contains a top-$\gamma_k$ configuration, such a configuration survives PRECHECK and the corresponding CAPSANDRUNS-thread in ICAR (Algorithm 1) ensures that $T$ is set to at most $2\text{OPT}_{\delta/2}^{\gamma_k}$ in Line 11 of Algorithm 1, that is, $T$ is continuously refined as new batches are evaluated. The number of configurations surviving PRECHECK can be bounded by looking at mean runtimes capped at the 0.35-quantile (upper bound on $p_i$). Together with the setting of $T$, this implies that at most a $\tilde{\mathcal{O}}(F(38\text{OPT}_{\delta/2}^{\gamma_{k+1}})$ fraction of the $|\mathcal{N}_k| = \tilde{\mathcal{O}}(1/\gamma_k)$ configurations survive PRECHECK. Considering that the number of runs carried out for each configuration is constant in PRECHECK, $\tilde{\mathcal{O}}(1/\delta)$ in the loop of Algorithm 1, and $\tilde{\mathcal{O}}(1/(\varepsilon^2\delta))$ in the last full CAR procedure, since the average runtime per configuration for $\mathcal{N}_k$ is $\text{OPT}_{\delta/2}^{\gamma_k}$ (by the analysis of CAR), the runtime bound of the theorem follows. Correctness (i.e., the fact that the procedure finds an $(\varepsilon, \delta, \gamma)$-optimal configuration) follows from that of CAR and because PRECHECK retains good configurations, as just shown.

## 4 Experiments

The basic setup and main results of our experimental analysis of ICAR are given below, while details are presented in Appendix D, along with a synthetic experiment examining ICAR's speedup as good configurations become increasingly rare. We compared against the best available configurators that come with theoretical guarantees. We used the improved version of CAR (CAR++), derived in this paper, which uses a smaller $b$-value than the original version, thanks to our improved analysis (see Section 3 and Appendix A for details). Including CAR++ in the experiments allowed us to separately examine the effects of two improvements we introduced: (i) the smaller number of samples $b$ needed in CAR, and (ii) the main conceptual innovation of this paper, the impatient discarding of configurations using PRECHECK. We attempted to compare against SPC [25] as well. However, in the experiments presented in Table 1, although SPC identified good configurations, it usually was not able to provide the required guarantees on $\varepsilon$ and $\delta$ even after running for twice as long as the slowest alternative considered (CAR): SPC did not provide guarantees for 7 out of the 9 scenarios while also being the slowest in the other two cases (1.56 and 1.91 times slower than CAR). Therefore, we decided not to include SPC in our further comparisons.

| | | Total CPU Time (days) | | | Number of Conf. Before/After PRECHECK | | | $R^\delta$ of returned conf. (secs) | | |
|---|---|---|---|---|---|---|---|---|---|---|
| | | $\gamma = 0.05$ | $\gamma = 0.02$ | $\gamma = 0.01$ | $\gamma = 0.05$ | $\gamma = 0.02$ | $\gamma = 0.01$ | $\gamma = 0.05$ | $\gamma = 0.02$ | $\gamma = 0.01$ |
| Minisat CNFuzzDD | ICAR | 101 (13) | 243 (15) | 467 (25) | 134 / 74 | 351 / 197 | 724 / 395 | 5.0 (0.1) | 4.9 (0.1) | 4.9 (0.1) |
| | CAR++ | **92 (5)** | **224 (16)** | **452 (18)** | 97 | 245 | 492 | 5.2 (0.1) | 4.9 (0.1) | 4.9 (0.1) |
| | CAR | 158 (18) | 368 (7) | 771 (22) | 97 | 245 | 492 | 5.2 (0.1) | 4.9 (0.1) | 4.9 (0.1) |
| CPLEX Regions200 | ICAR | **164 (91)** | **275 (101)** | **420 (103)** | 134 / 10 | 351 / 15 | 724 / 26 | 34.8 (4.3) | 29.8 (2.2) | 28.5 (1.8) |
| | CAR++ | 229 (20) | 567 (28) | 1098 (88) | 97 | 245 | 492 | 35.3 (4.3) | 32.0 (2.2) | 29.8 (1.8) |
| | CAR | 524 (53) | 1295 (64) | 2549 (199) | 97 | 245 | 492 | 35.3 (4.5) | 31.9 (1.6) | 29.8 (2.2) |
| CPLEX RCW | ICAR | **1284 (391)** | **2030 (302)** | **4072 (239)** | 134 / 18 | 351 / 44 | 724 / 97 | 156.1 (11.9) | 146.5 (4.1) | 143.3 (4.9) |
| | CAR++ | 1728 (375) | 3644 (185) | 7526 (131) | 97 | 245 | 492 | 162.1 (11.9) | 149.1 (4.1) | 143.3 (4.9) |
| | CAR | 3306 (502) | 7591 (192) | 15658 (258) | 97 | 245 | 492 | 160.1 (13.3) | 149.1 (4.7) | 143.3 (4.9) |

Table 1: Total CPU time in days to find a $(0.05, 0.1, \gamma)$-optimal configuration, the number of configurations before and after PRECHECK, and the quality of the returned configurations, as measured by $\delta$-capped mean runtime with $\delta = 0.1$. For CAR and CAR++, the number of configurations sampled is reported. Error terms (in parentheses) are standard deviations over five runs.

**Datasets.** We looked at two datasets from MIP and one from SAT. We considered true runtime data from the `minisat` SAT solver on instances generated by CNFuzzDD (http://fmv.jku.at/cnfuzzdd), which was examined in past work [35, 36, 25]. For the MIP scenarios, we looked at the CPLEX integer program solver on combinatorial auction instances (Regions200 [27]) and problems from wildlife conservation (RCW [1]). To generate sufficient MIP runtime data, following Hutter et al. [23], we used an *Empirical Performance Model (EPM)*—a random forest model trained on existing runtime data—to predict the runtime of new configurations on new instances. EPMs can do surprisingly well at predicting individual runtimes, particularly on the MIP datasets we consider. More importantly for our purposes, Eggensperger et al. [13] showed that such EPMs are effective surrogates for algorithm configuration, capturing key properties of runtime distributions such as the relative quality of configurations. We note that similar surrogates have also been used to guide search procedures [20, 9, 34, 38], to build algorithm portfolios [32, 37], to impute missing data [10], and to optimize hyperparameters from limited observations [33].

**Main Results.** Table 1 shows the total CPU time needed to find a $(0.05, 0.1, \gamma)$-optimal configuration on each dataset with the same total failure probability $(0.05)$ and with different values of $\gamma$. The parameters were not specifically chosen; results for varying $\varepsilon$ and $\delta$ are reported in Appendix D. ICAR consistently outperformed CAR in all cases; ICAR outperformed CAR++ on the MIP datasets and was competitive on the SAT one. The performance improvement was largest when the PRECHECK mechanism managed to discard the most configurations; the MIP datasets have relatively more weak configurations, enabling PRECHECK to filter out more configurations quickly (see Fig. 2 in the Appendix for the distribution of configuration means). When $\gamma$ is relatively small, ICAR was more likely to sample a really good configuration, making it easier to discard weak ones. In this case its runtime was as little as half that of CAR++, a significant improvement. Despite taking less total CPU time, ICAR actually sampled *more* configurations than CAR did. To understand this phenomenon better, Fig. 1 shows the time spent running each configuration. For all datasets the plots nearly overlap for the very best few configurations, indicating that ICAR treated these good configurations in much the same way as CAR or CAR++. However, the effect of the PRECHECK mechanism is clear, as ICAR ran many bad configurations for near-zero time, discarding them quickly. In cases where a bad configuration made it past PRECHECK (largest spikes in the blue curve), ICAR ran it for an amount of time similar to CAR++. Finally, the empirical mean $\delta$-capped runtime ($R^\delta$) of the returned configuration is reported in Table 1. All configurators returned solutions with similar quality, but thanks to its ability to examine more configurations, ICAR often did slightly better.

## 5 Conclusions

This paper presented a novel algorithm configuration method, ICAR, that selects configurations from an infinite pool with optimal theoretical guarantees up to logarithmic factors under mild conditions. While earlier theoretically grounded methods thoroughly test all configurations, ICAR—like successful heuristic approaches—quickly discards less promising ones. As a result, ICAR achieves significant speedups, particularly in needle-in-a-haystack scenarios. It thus constitutes an important step towards closing the gap between theoretical and heuristic procedures.

A key limitation is that our work focuses simply on evaluating randomly sampled configurations. We do note that state-of-the-art heuristic methods also evaluate many random configurations to avoid getting stuck in local optima, so analyzing such procedures is of obvious practical importance.

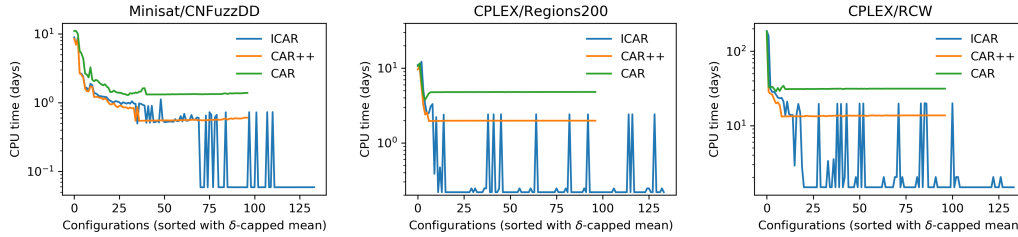

Figure 1: CPU time spent on each configuration while searching for a $(0.05, 0.1, 0.05)$-optimal one (note the log scale on the $y$-axis). CAR and CAR++ allocated a significant amount of time to evaluating bad configurations, while ICAR discarded many of these with near minimal work via its PRECHECK routine. The large spikes in the ICAR curve are those configurations that fail to be rejected by the first call to PRECHECK. Smaller spikes are configurations that were also rejected by PRECHECK, but the decision took more time (e.g., $T$ was larger in PRECHECK or the configuration was rejected in the second phase of PRECHECK).

Furthermore, ICAR can be understood as a way of weighing different candidate configurations against each other, which could be proposed by model- or gradient-based methods as well as by random sampling (see, e.g., an argument to this effect in [24, Theorem 7.1]).

## Broader Impact

We expect that our theorems will guide the design of future algorithm configuration procedures. We note that speeding up computationally expensive algorithms saves time, money, and electricity, arguably reducing carbon emissions and yielding social benefit. The algorithms we study can be be applied to a limitless range of problems and so could yield both positive and negative impacts; however, we do not foresee our work particularly amplifying such impacts beyond the computational speedups already discussed.

## Acknowledgments and Disclosure of Funding

This work was supported by Compute Canada, NSERC Discovery Grants, a DND/NSERC Discovery Grant Supplement, CIFAR Canada AI Research Chair grants at the Alberta Machine Intelligence Institute, and DARPA award FA8750-19-2-0222, CFDA #12.910, sponsored by the Air Force Research Laboratory.

## Footnotes

[1]As usual, we treat the cumulative runtime of all the configurations tried as the total search time. One could also consider including the overhead imposed by the configuration algorithm itself. However, beyond being difficult to model, this cost is typically negligible compared to the runtime of the configurations.

[2]We can see $\Pi$ as reflecting beliefs about the distribution of good configurations in the parameter space. This implicitly neglects any search procedure that leverages structural assumptions about the parameter space.

[3]With a slight abuse of terminology, throughout we use the same expression for capping with timeouts ($\tau$) and quantiles ($\delta$), when the interpretation is clear from the context; we specify the type of capping otherwise.

[4]We use "total work" and "total runtime" interchangeably; both sum over all parallel threads.

[5]We use the standard $\mathcal{O}$ and $\tilde{\mathcal{O}}$ notation, where the latter hides poly-logarithmic factors.

[6]Essentially this holds since we need $\tilde{\Omega}(\frac{1}{\varepsilon^2\delta})$ sample runs to estimate the $\delta$-capped runtime of a configuration with accuracy $\varepsilon$, as the maximum runtime for configuration $i$ on some instance can be as large as $R_\delta(i)/\delta$.

[7]Almost all guarantees provided in this paper are based on random sampling and hence hold with high probability. For brevity, when it is clear from the context, we often omit the 'high-probability' qualifier.

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
