[Supplementary Material]

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

[8]This constant 0.35 can be set arbitrarily, and it only affects other constants in the algorithm. It was set to 0.35 so that these constant do not increase beyond how large they have to be to guarantee other statements with high probability.

[9]In fact, OPT can even be infinite while $\text{OPT}_{\frac{\delta}{2}}$ is still finite.

[10]For precise definitions, the reader is referred to the original paper Weisz et al. [36].

[11]http://www.cs.ubc.ca/labs/beta/Projects/EPMs/

[12]https://www.ml4aad.org/automated-algorithm-design/performance-prediction/epms/

[13]https://www.cs.ubc.ca/~drgraham/datasets.html

[14]https://github.com/empennage98/icar

[15]http://fmv.jku.at/cnfuzzdd/

[16]https://github.com/deepmind/leaps-and-bounds

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

# A  Proof of Theorem 1

First we give a detailed outline of the proof, followed by the actual statements and their proofs.

The first step of the proof improves the analysis of CAPSANDRUNS given in [36]. In [36], the value of $b$ was $\left\lceil \frac{48}{\delta} \log\left(\frac{3n}{\zeta}\right)\right\rceil$, which we replace here with $\left\lceil \frac{26}{\delta} \log\left(\frac{2n}{\zeta}\right)\right\rceil$. This value is used in the original analysis of CAPSANDRUNS twice, in Lemma 2 and Lemma 3 of [36]. The analysis of Lemma 2 still holds with the new value without any change, while we give a new proof for Lemma 3 of [36]: the difference is that in the new proof we use the Bernstein inequality (Appendix E) rather than its empirical version. The new version of the lemma, Lemma 2, is slightly stronger, which means we can replace $2Tb$ with $1.5Tb$ in Line 3 of the sub-routine QUANTILEEST. Note that this change of the value of $b$ itself improves the runtime of CAR, and we call the resulting algorithm CAR++, which will also be examined in the experiment section.

To prove Theorem 1, we need to (i) prove the correctness of IMPATIENTCAPSANDRUNS, that is, the $(\varepsilon, \delta, \gamma)$-optimality of the configuration returned by the algorithm; and (ii) give a bound on the total runtime. Starting with the correctness, we note that the algorithm proceeds in iterations from $K - 1$ to 0 in decreasing order, sampling bigger and bigger sets of configurations $\mathcal{N}_k$. Each new set $\mathcal{N}_k$, together with those configurations sampled before for $k' > k$, contains an $\mathrm{OPT}_{\delta/2}^{\gamma_k}$-optimal configuration with high probability (Lemma 4), in other words, a configuration from an exponentially decreasingly small quantile of the best configurations. The size of $\mathcal{N}_k$, for all $k \in [0, K - 1]$ is roughly $\log(K/\zeta)/\gamma_k$ (Lemma 6). Next, we prove in Lemma 8 that PRECHECK does not reject a good configuration, and does reject a truly bad configuration. Unlike other parts of the proof, we do not guarantee this to hold with high probability for all configurations, instead we guarantee it to hold with high probability for any one configuration per each iteration $k$; this will be chosen later to be one of the $\mathrm{OPT}_{\delta/2}^{\gamma_k}$-optimal configurations. Then, Lemma 10 shows that there remains an $\mathrm{OPT}_{\delta/2}^{\gamma_k}$-optimal configuration after each iteration $k$ (Line 11 of Algorithm 1) that is not rejected by QUANTILEEST or RUNTIMEEST. This is because even if our designated configuration was rejected by PRECHECK, that means that there was an even better configuration, which from the proof of CAPSANDRUNS, by Lemma 9, will not be rejected by QUANTILEEST or RUNTIMEEST. Several corollaries follow from this. Corollary 11 shows that with high probability, the configuration IMPATIENTCAPSANDRUNS returns in the end is $(\varepsilon, \delta, \gamma)$-optimal, showing the correctness of the algorithm To prove the runtime bound, we start by showing that in every iteration $k$, $T$ is set to at most $2\mathrm{OPT}_{\delta/2}^{\gamma_k}$, after evaluating a configuration for no more than $4b\mathrm{OPT}_{\delta/2}^{\gamma_k}$ time (Corollary 12). From this, Corollary 13 deduces a runtime bound for CAR in each iteration, which depends on the number of configurations surviving PRECHECK. Using the correctness analysis of PRECHECK (Lemma 8), Lemma 14 gives an upper bound on this number, essentially saying that roughly only $F(38\mathrm{OPT}_{\delta/2}^{\gamma_{k+1}})$ fraction of the $\mathcal{N}_k$ configurations survive PRECHECK in round $k$, where $F(x)$ is roughly the probability of a random configuration sampled from $\Pi$ having a larger 0.35th quantile-capped[8] runtime than $x$. That is, essentially only those configurations survive which can solve at least 65% of the problem instances reasonably fast.

This is complemented by Lemma 15, which gives a runtime bound for PRECHECK, relying on Lines 13 and 21 of PRECHECK (Algorithm 4) stopping lengthy evaluations. To finish the proof, we combine the runtime bounds for all the components of ICAR discussed above. The lemmas above introduce various high-probability events under which their statements hold (by guaranteeing mostly that our bounds on the runtime caps and on the average runtimes hold), and a union bound over them proves that all those events hold simultaneously with probability at least $1 - 12\zeta$, proving Theorem 1.

## A.1  Formal Proof

**Lemma 2** (Improved version of Lemma 3 of [36])**.** *Let $\tau$ be a constant satisfying $0 \leq \tau \leq t_{\delta/2}(i)$, and let $Z_\tau(i, j)$, $j \in [1, b]$, be $b$ runtime measurements of configuration $i$ with timeout $\tau$. Let $\bar{Z}_\tau(i)$ be their average and $R_\tau(i)$ their expectation. Then for $c > 0$, $\Pr\left(|\bar{Z}_\tau(i) - R_\tau(i)| \geq cR_\tau(i)\right) \leq$*

$2\exp\left(\frac{b\delta c^2}{4(1+c/3)}\right)$. In particular, for $S_i = \{\frac{1}{2}R_\tau(i) \leq \bar{Z}_\tau(i) \leq 1.5R_\tau(i)\}$ and $b = \left\lceil \frac{26}{\delta}\log\left(\frac{2n}{\zeta}\right)\right\rceil$, we have $\Pr(S_i^c) \leq \frac{\zeta}{n}$ (by substituting $c = \frac{1}{2}$).

*Proof.* Since $Z_\tau(i,j) \leq \tau$, $\mathrm{Var}(Z_\tau(i)) = \mathrm{Var}_{j\sim\Gamma}[Z_\tau(i,j)] \leq \mathbb{E}_{j\sim\Gamma}Z_\tau^2(i,j) \leq \tau R_\tau(i)$. As at least $\delta/2$ fraction of instances run longer than $\tau$, we have that $R_\tau(i) \geq \frac{\delta}{2}\tau$, so $\mathrm{Var}(Z_\tau(i)) \leq \frac{2}{\delta}R_\tau^2(i)$. Using the Bernstein inequality,

$$\Pr\left(|\bar{Z}_\tau(i) - R_\tau(i)| \geq cR_\tau(i)\right) \leq 2\exp\left(-\frac{bc^2 R_\tau^2(i)/2}{\frac{1}{3}\tau cR_\tau(i) + \mathrm{Var}(Z_\tau(i))}\right)$$

$$\leq 2\exp\left(-\frac{bc^2 R_\tau^2(i)/2}{\frac{2}{3\delta}cR_\tau^2(i) + \frac{2}{\delta}R_\tau^2(i)}\right)$$

$$= 2\exp\left(\frac{b\delta c^2}{4(1+c/3)}\right).$$

$\square$

**Remark 3.** *From Lemma 6 of [36], there is an event $E_1$ (with the notation of [36], this event is $E_1 \cap E_2 \cap E_3$) with $\Pr(E_1) \geq 1 - 6\zeta$, under which all the high-probability statements in the analysis of* CAPSANDRUNS *hold for the algorithm with the constants improved as above. Note that throughout ICAR, we only run one CAR, but pause and resume its threads as we go through the iterations $k \in [-1, K-1]$. Pausing and resuming threads has no effect on the correctness proof of CAR, so throughout the execution of ICAR, all the high-probability statements that hold for CAR also hold for ICAR. In particular, $E_1$ guarantees that the average runtime estimates of CAR (used in* RUNTIMEEST *and* QUANTILEEST*) are close to their expectations, and that* QUANTILEEST *measures an accurate cap for each configuration such that $t_\delta(i) \leq \tau_i \leq t_{\delta/2}(i)$.*

**Lemma 4.** *There is an event $E_2$ with $\Pr(E_2) \geq 1 - \zeta$ such that under $E_2$, for all integers $k \in [0, K-1]$, there is a configuration $i \in \bigcup_{j=k}^{K-1}\mathcal{N}_j$ with $R^{\frac{\delta}{2}}(i) \leq \mathrm{OPT}_{\delta/2}^{\gamma_k}$ after Line 2 of* IMPATIENTCAPSANDRUNS *(Algorithm 1).*

*Proof.* For any $i$ chosen randomly from the distribution $\Pi$, $R^{\frac{\delta}{2}}(i) \leq \mathrm{OPT}_{\delta/2}^{\gamma_k}$ with probability at least $\gamma_k$. As the configurations are sampled independently, the probability that none of the sampled configurations are optimal for $\gamma_k$ is at most $(1-\gamma_k)^{|\bigcup_{j=k}^{K-1}\mathcal{N}_j|} = (1-\gamma_k)^{\left\lceil \frac{\log(\zeta/K)}{\log(1-\gamma_k)}\right\rceil} \leq \zeta/K$. Applying the union bound over $k \in [0, K-1]$, with probability at least $1 - \zeta$, for all $k \in [0, K-1]$, there is a configuration $i$ with $R^{\frac{\delta}{2}}(i) \leq \mathrm{OPT}_{\delta/2}^{\gamma_k}$ sampled into $\bigcup_{j=k}^{K-1}\mathcal{N}_j$ at Line 2 of IMPATIENTCAPSANDRUNS. $\square$

**Remark 5.** *Noting that $\gamma_0 = \gamma$, and $\bigcup_{k=0}^{K-1}\mathcal{N}_k = \mathcal{N}$, the previous lemma with $k = 0$ states that under $E_2$, a configuration $i$ with $R^{\frac{\delta}{2}}(i) \leq \mathrm{OPT}_{\delta/2}^{\gamma}$ is sampled into $\mathcal{N}$.*

In the following we refer to the last part of Algorithm 1 (Lines 14 to 18) iteration $-1$, denote it with $k = -1$, and accordingly define $\overline{\mathcal{N}}_{-1} = \overline{\mathcal{N}}$ and $\mathcal{N}_{-1} = \mathcal{N}$.

**Lemma 6.** *After Line 2 of Algorithm 1, for all $k \in [-1, K-1]$, $|\mathcal{N}_k| \leq \log(K/\zeta)/\gamma_k + 1$.*

*Proof.* Using that for any $x \in (0,1)$, $x \leq -\log(1-x)$, we have for any $k \in [-1, K-1]$ that

$$|\mathcal{N}_k| \leq \left\lceil \frac{\log(\zeta/K)}{\log(1-\gamma_k)}\right\rceil \leq \frac{\log(K/\zeta)}{-\log(1-\gamma_k)} + 1 \leq \log(K/\zeta)/\gamma_k + 1.$$

$\square$

By [36, Lemma 2], under $E_1$, and noting that $T$ can only be set by RUNTIMEEST evaluating a configuration after the cap $\tau$ for that configuration has already been measured by QUANTILEEST, we immediately obtain the following:

**Lemma 7.** *If a configuration $i$ sets $T$, then $t_\delta(i) \leq \tau_i \leq t_{\delta/2}(i)$.*

The next lemma gives conditions for configurations being accepted or rejected by PRECHECK.

**Lemma 8.** *Suppose $\delta \leq 0.2$ and assume that we are in the PRECHECK call in iteration $-1 \leq k < K - 1$ of Algorithm 1 (recall that iteration -1 refers to the last part of the algorithm after the iteration loop is finished). Let $\Pr_k$ denote the conditional probability conditioned on all the random events before the call to PRECHECK. Let $i'$ be the configuration that was last evaluated to set $T$ by RUNTIMEEST (in an iteration $k' > k$). For any $i \in \mathcal{M}$, there is an event $E_{3,k,i}$ with $\Pr_k(E_{3,k,i}) \geq 1 - 4\zeta/K$ such that under $E_1$ and $E_{3,i}$, (1) if $R^{\frac{\delta}{2}}(i) \leq R_{\tau_{i'}}(i')$, $i$ will not be rejected by PRECHECK, and (2) if $R^{0.35}(i) \geq 19T$, then $i$ will be rejected by PRECHECK.*

*Proof.* Consider the evaluation of configuration $i$ in PRECHECK. Let $I_l$ be the indicator that the $l^{th}$ instance in Phase I of PRECHECK takes at least $t_{1/10}(i)$ time to complete (without capping). The $I_l$ are independent and identically distributed Bernoulli random variables with $\Pr_k(I_l = 1) = 1/10$. We use the Chernoff bound (Appendix E) to get that $\Pr_k\left(\sum_{l=0}^{b'} I_l > 0.2b'\right) = \Pr_k\left(\sum_{l=0}^{b'} I_l > \frac{1}{10}b'(1+1)\right) \leq \exp\left(-\frac{1}{30}b'\right) \leq \zeta/(2K)$. Let $E_{3,k,i,1}$ be the event that $\sum_{l=0}^{b'} I_l \leq 0.2b'$. Similarly, defining $J_l$ to be the indicator that the $l^{th}$ instance in Phase I of PRECHECK takes at least $t_{0.35}$ time to complete (without capping), noting that $\Pr_k(J_l = 1) = 0.35$, the Chernoff bound implies that $\Pr_k\left(\sum_{l=0}^{b'} J_l < 0.2b'\right) = \Pr_k\left(\sum_{l=0}^{b'} J_l > 0.35b'(1 - \frac{0.35-0.2}{0.35})\right) \leq \zeta/(2K)$. Let $E_{3,k,i,2}$ be the event that $\sum_{l=0}^{b'} J_l \geq 0.2b'$.

So for any configuration $i \in \mathcal{M}$, under $E_{3,k,i,1}$, the number of samples from the $1/10$-tail will be at most $\lfloor 0.2b' \rfloor$, and under $E_{3,k,i,2}$, the number of samples from the $0.35$-tail will be at least $\lceil 0.2b' \rceil$, so picking the $\lceil 0.8b' \rceil^{th}$ finished run and denoting it by $\tau'$ (in Line 15 of Algorithm 4) ensures under and $E_{3,k,i,1}$ and $E_{3,k,i,2}$ that $t_{0.35}(i) \leq \tau' \leq t_{1/10}(i)$. For $\delta \leq 0.2$ (which we have by assumption), this implies that $\tau' \leq t_{\delta/2}(i)$.

By [36, Lemma 7], under $E_1$, $R_{\tau_{i'}}(i') \leq T$. Assuming that $R^{\frac{\delta}{2}}(i) \leq R_{\tau_{i'}}(i')$ for (1), we have $R^{\frac{\delta}{2}}(i) \leq T$. Then under $E_{3,k,i,1}$, $R_{\tau'}(i) \leq R^{1/10}(i) \leq T$ (as by assumption $\delta \leq 0.2$ and $\tau' \leq t_{1/10}(i)$, so $R_{\tau'}(i) \leq R^{1/10}(i) \leq R^{\frac{\delta}{2}}(i) \leq R_{\tau_{i'}}(i') \leq T$). There are two cases in which we reject configuration $i$. First, if $\text{avg}(Y) - C \geq T$. By the empirical Bernstein bound [4] (Appendix E), there is an event $E_{3,k,i,3}$ such that $\Pr_k(E_{3,k,i,3}) \geq 1 - \zeta/K$, and under $E_{3,k,i,3}$, $|\text{avg}(Y) - \mathbb{E}_k[\text{avg}(Y)|\tau']| \leq C$. As $\mathbb{E}_k[\text{avg}(Y)|\tau'] = R_{\tau'}(i) \leq T$, we have that under $E_1$, $E_{3,k,i,1}$ and $E_{3,k,i,3}$, configuration $i$ in iteration $k$ will not be rejected in Line 27 of PRECHECK if (1) holds.

The second type of rejection happens in Line 13 of PRECHECK when Phase I of PRECHECK runs for at least $1.9Tb'$ time. For each run $l$ that is performed in Phase I, denote by $X_l$ the hypothetical runtime of that instance if the cap were $t_{1/10}(i)$, and by $Y_l$ the run if the runtime cap were $t_{0.35}(i)$. From the above, under $E_{3,k,i,1}$ $\tau' \leq t_{1/10}(i)$, and if we had to abort then that means we haven't run any instance for $\tau'$ time yet, so by denoting measurements performed so far by Phase I of PRECHECK by $\bar{X}_l$, we have $\bar{X}_l \leq X_l$, so when we abort we have that $\text{avg}(X) \geq 1.9T$.

Applying Lemma 2 with $c = 0.9$, $b = b' = \left\lceil 32.1 \log\left(\frac{2K}{\zeta}\right) \right\rceil$, $\delta = 0.2$, and $\tau = t_{1/10}(i)$, we get that $\Pr_k\left(|\text{avg}(X) - R^{1/10}(i)| \geq 0.9R^{1/10}(i)\right) \leq 2\exp\left(b'\frac{81}{5 \cdot 520}\right) \leq \frac{\zeta}{K}$. Denote by $E_{3,k,i,4}$ the event that $\text{avg}(X) \leq 1.9T$. Then, for (1), under $E_1$ and $E_{3,k,i,1}$, by the above $R^{1/10}(i) \leq T$, we have $\Pr_k(\text{avg}(X) \geq 1.9T) \leq \Pr_k(\text{avg}(X) - R^{1/10}(i) > 0.9R^{1/10}(i)) \leq \frac{\zeta}{K}$, so $\Pr_k(E_{3,k,i,4}) \geq 1 - \frac{\zeta}{K}$, and under $E_1$ and $E_{3,k,i,4}$, this configuration will not be rejected in Line 13 of PRECHECK if it satisfies (1).

For (2), let $E_{3,k,i,5}$ the event that $\text{avg}(Y) \geq 0.1R^{0.35}(i)$. Apply Lemma 2 with the same parameters except $\tau = t_{0.35}(i)$, to get that $\Pr_k(\text{avg}(Y) \leq 0.1R^{0.35}(i)) = \Pr_k(R^{0.35}(i) - \text{avg}(Y) \geq 0.9R^{0.35}(i)) \leq \Pr_k(|\text{avg}(Y) - R^{0.35}(i)| \geq 0.9R^{0.35}(i)) \leq 2\exp\left(b'\frac{81}{5 \cdot 520}\right) \leq \frac{\zeta}{K}$. For PRECHECK to not reject a configuration $i$, it measures a cap $\tau' \geq t_{0.35}(i)$ (under $E_{3,k,i,2}$), and so the measurements $\bar{Y}_l$ satisfy $\bar{Y}_l \geq Y_l$, so we spend $b'\text{avg}(\bar{Y}_l) \geq b'\text{avg}(Y_l) \geq 0.1R^{0.35}(i)$ time for configuration $i$ under $E_{3,k,i,5}$. Thus, with probability at least $\Pr_k(E_{3,k,i,4}) \geq 1 - \zeta/K$, a configuration where $R^{0.35}(i) \geq 19T$ is rejected in Line 13. Taking a union bound and letting

$E_{3,k,i} = E_{3,k,i,1} \cap E_{3,k,2} \cap E_{3,k,i,3} \cap E_{3,k,i,4} \cap E_{3,k,i,5}$ (the event that all the high-probability statements above hold for configuration $i$ and iteration $k$), we have that $\Pr_k(E_{3,k,i}) \geq 1 - 4\zeta/K$. $\quad\square$

From the proof of [36, Theorem 1] we can extract the following result:

**Lemma 9.** *Let $\mathcal{N}$ be the set of configurations* CAPSANDRUNS *is called with, and $\mathcal{N}'$ the ones among these that are not rejected in* QUANTILEEST. *Let $i_* = \min_{i \in \mathcal{N}'} R_{\tau_i}(i)$. Under $E_1$, $i_*$ is not rejected in* RUNTIMEEST *and* CAPSANDRUNS *returns a configuration $I$ for which $R_{\tau_I}(I) \leq (1+\varepsilon) R_{\tau_{i_*}}(i_*)$.*

To proceed, we instantiate the events $E_{3,k,i}$ of Lemma 8 for one of the best configurations $i$ in $\mathcal{N}_k$. By Lemma 4 and Remark 5, under $E_2$, for every iteration $k$ of IMPATIENTCAPSANDRUNS, there is a configuration $\hat{\imath}_k^* \in \bigcup_{j=k}^{K-1} \mathcal{N}_j$ such that $R^{\frac{\delta}{2}}(\hat{\imath}_k^*) \leq \mathrm{OPT}_{\delta/2}^{\gamma_k}$. Furthermore, this guarantees that $\hat{\imath}_0^* \in \mathcal{N}$ satisfies $R^{\frac{\delta}{2}}(\hat{\imath}_0^*) \leq \mathrm{OPT}_{\delta/2}^{\gamma}$, which also implies, through the first part of Lemma 9, that under $E_1$, there is a configuration $\hat{\imath}_{-1}^*$ satisfying $R^{\frac{\delta}{2}}(\hat{\imath}_{-1}^*) \leq \mathrm{OPT}_{\delta/2}^{\gamma}$. Now we define the following event $E_4 \subset E_1 \cap E_2$ as $E_4 = \cap_{k=-1}^{K-2} E_{3,k,\hat{\imath}_k^*} \cap E_1 \cap E_2$.

**Lemma 10.** $\Pr(E_4) \geq 1 - 11\zeta$. *Under $E_4$, for all integer $0 \leq k \leq K-1$, there is a configuration $i_k^*$ remaining in $\bigcup_{j=k}^{K-1} \overline{\mathcal{N}}_j$ at the end of the $k^{th}$ iteration (after Line 11 in Algorithm 1), that is not rejected by* QUANTILEEST *or* RUNTIMEEST, *for which $R_{\tau_{i_k^*}}(i_k^*) \leq \mathrm{OPT}_{\delta/2}^{\gamma_k}$. Similarly, there is a configuration $i_*$ remaining in $\overline{\mathcal{N}}$ at the end of the final* CAPSANDRUNS *call (after Line 17 in Algorithm 1), for which $R_{\tau_{i_*}}(i_*) \leq \mathrm{OPT}_{\delta/2}^{\gamma}$.*

*Proof.* By the union bound, taking also into account the lower bounds on the probabilities of $E_1$, $E_2$, and $E_{3,k,i_k^*,j}$ (given by Remark 3, Lemma 4, Lemma 8, Lemma 14), we have $\Pr(E_1 \cap E_2) \geq 1 - 7\zeta$, and $\Pr(E_4) \geq 1 - 11\zeta$.

Now suppose $E_4$ holds (this also means that $E_1$ and $E_2$ hold). Let $i'$ denote the configuration that last set $T$.

For $k = K-1$ there is no PRECHECK as $T = \infty$, in other words nothing is rejected by PRECHECK. For iterations $0 \leq k \leq K-2$, and for the final CAPSANDRUNS call, either $R^{\frac{\delta}{2}}(\hat{\imath}_k^*) \leq R_{\tau_{i'}}(i')$, in which case by Lemma 8, under $E_1$ and $E_{3,k,\hat{\imath}_k^*}$, $\hat{\imath}_k^*$ is not rejected, or $R^{\frac{\delta}{2}}(\hat{\imath}_k^*) > R_{\tau_{i'}}(i')$. Thus under $E_4$, $R^{\frac{\delta}{2}}(\hat{\imath}_k^*) > R_{\tau_{i'}}(i')$ holds whenever $\hat{\imath}_k^*$ is rejected. We assume this for the rest of the proof.

The remainder of this proof handles iterations $0 \leq k \leq K-1$, but the arguments transfer for the final CAPSANDRUNS call case by writing $\gamma$ and $\hat{\imath}_{-1}^*$ instead of $\gamma_k$ and $\hat{\imath}_k^*$. We investigate the two possible cases:

- If $\hat{\imath}_k^*$ is not rejected by PRECHECK, then under $E_1$ by Lemma 8 in [36], there is an $i$ in the set of configurations CAPSANDRUNS is called with, that will not be rejected by QUANTILEEST, and for which $R_{\tau_i}(i) \leq R^{\frac{\delta}{2}}(\hat{\imath}_k^*) \leq \mathrm{OPT}_{\delta/2}^{\gamma_k}$.

- If $\hat{\imath}_k^*$ is rejected by PRECHECK, then $i'$ is a configuration not rejected by QUANTILEEST (as it set $T$), for which $R_{\tau_{i'}}(i') \leq R^{\frac{\delta}{2}}(\hat{\imath}_k^*) \leq \mathrm{OPT}_{\delta/2}^{\gamma_k}$.

In either case, there is a configuration $i$ not rejected by QUANTILEEST, for which $R_{\tau_i}(i) \leq \mathrm{OPT}_{\delta/2}^{\gamma_k}$. Thus by Lemma 9, under $E_1$, there is a configuration $i_k^*$ not rejected by QUANTILEEST or RUNTIMEEST for which $R_{\tau_{i_k^*}}(i_k^*) \leq R_{\tau_i}(i) \leq \mathrm{OPT}_{\delta/2}^{\gamma_k}$. $\quad\square$

**Corollary 11.** *Under $E_4$, the configuration returned by* IMPATIENTCAPSANDRUNS *is $(\varepsilon, \delta, \gamma)$-optimal.*

*Proof.* By Lemma 9 and Lemma 10, under $E_4$, the final CAPSANDRUNS call returns with a configuration $I$ for which $R_{\tau_I}(I) \leq (1+\varepsilon) R_{\tau_{i_*}}(i_*) \leq (1+\varepsilon)\mathrm{OPT}_{\delta/2}^{\gamma}$. Under $E_1$, $R^{\delta}(I) \leq R_{\tau_I}(I)$, so $I$ is $(\varepsilon, \delta, \gamma)$-optimal. $\quad\square$

**Corollary 12.** *Under $E_4$, for all iterations $0 \leq k \leq K - 1$, $T$ is set by* QUANTILEEST *to at most* $2\text{OPT}_{\delta/2}^{\gamma_k}$, *and the combined time spent by* QUANTILEEST *and* RUNTIMEEST *evaluating the configuration that has set $T$ is bounded by* $4b\text{OPT}_{\delta/2}^{\gamma_k}$ *when it sets $T$.*

*Proof.* Take $i_k^*$ as in Lemma 10. Since $i_k^*$ is not rejected in either QUANTILEEST or RUNTIMEEST, its $b$ measurements in RUNTIMEEST will complete, and by [36, Lemma 4], under $E_1$, this measurement will be at most $2R_{\tau_{i_k^*}}(i_k^*) \leq 2\text{OPT}_{\delta/2}^{\gamma_k}$. $T$ is thus set to at most this value. From the proof of [36, Lemma 5], the work spent by QUANTILEEST and RUNTIMEEST evaluating $i_k^*$ is bounded by $4b\text{OPT}_{\delta/2}^{\gamma_k}$ time. $\square$

**Corollary 13.** *Suppose $E_4$ holds. Then for all iterations $0 \leq k \leq K - 1$,* CAPSANDRUNS *performs at most* $\tilde{\mathcal{O}}\left(b|\overline{\mathcal{N}}_k|\text{OPT}_{\delta/2}^{\gamma_k}\right)$ *work.*

*Proof.* By Corollary 12, under $E_4$, in iteration $k$, $T$ is set to at most $2\text{OPT}_{\delta/2}^{\gamma_k}$, after which by the proof of [36, Lemma 5], each configuration performs at most $\tilde{\mathcal{O}}\left(b\text{OPT}_{\delta/2}^{\gamma_k}\right)$ work. Also by Corollary 12, the work performed by the configuration that set $T$ to this value in iteration $k$ is upper bounded by $4b\text{OPT}_{\delta/2}^{\gamma_k}$. Since configurations are run in parallel, all the other configurations performed at most this amount of work in the meantime. Thus in total CAPSANDRUNS performs at most $\tilde{\mathcal{O}}\left(b|\overline{\mathcal{N}}_k|\text{OPT}_{\delta/2}^{\gamma_k}\right)$ work in iteration $k$. $\square$

**Lemma 14.** *There is an event $E_5$ such that $\Pr(E_5) \geq 1 - \zeta$, and under $E_5$, $E_1$, and $E_4$, for all iterations $k \in [-1, K - 2]$, the number of configurations not rejected by* PRECHECK *can be bounded as*

$$|\overline{\mathcal{N}}_k| \leq (\log(K/\zeta) + 1)\left[F(38\text{OPT}_{\delta/2}^{\gamma_{k+1}})\frac{1}{\gamma_k} + \sqrt{2F(38\text{OPT}_{\delta/2}^{\gamma_{k+1}})\frac{1}{\gamma_k}} + \frac{2}{3}\right],$$

*where $F(x) = \Pr_{i \sim \Pi}(R^{0.35}(i) \leq x) + 4\zeta/K$.*

*Proof.* Note that for the first call of PRECHECK, with $k = K - 1$, PRECHECK returns its input without any modification, so $|\mathcal{M}'| = |\mathcal{M}|$. For the rest of the calls, $-1 \leq k < K - 1$.

Denoting by $B_i$ the indicator whether configuration $i \in \mathcal{N}_k$ is accepted by PRECHECK. Since elements of $\mathcal{N}_k$ are independent and identically distributed random variables, and there are no interactions between configurations being evaluated by PRECHECK, the outcomes $B_i$ of PRECHECK are also independent and identically distributed. By Lemma 8, under $E_1$, PRECHECK rejects a configuration $i$ if $R^{0.35}(i) \geq 19T$ with probability at least $1 - 4\zeta/K$, so the probability of reject is at least $\Pr_{i \sim \Pi}(R^{0.35}(i) \geq 19T \,|\, T)(1 - 4\zeta/K) \geq \Pr_{i \sim \Pi}(R^{0.35}(i) \geq 19T \,|\, T) - 4\zeta/K = 1 - F(19T)$, so the conditional probability of accept is at most $F(19T)$. The number of configurations not accepted is $\sum_{i \in \mathcal{N}_k} B_i$. Let $E_{5,k}$ be the event that $\sum_{i \in \mathcal{N}_k} B_i \leq F(19T)|\mathcal{N}_k| + \sqrt{2F(19T)|\mathcal{N}_k|\log\frac{K}{\zeta}} + \frac{2}{3}\log\left(\frac{K}{\zeta}\right)$. By the Bernstein inequality, $\Pr\left(E_{5,k}^c \,\middle|\, T\right) \leq \frac{\zeta}{K}$. Since this holds for all values of $T$, we have that $\Pr\left(E_{5,k}^c\right) \leq \frac{\zeta}{K}$, so by a union bound over $k \in [-1, K - 2]$, $\Pr(E_5) \geq 1 - \zeta$ for the event $E_5 = \bigcap_{k=-1}^{K-2} E_{5,k}$. By Lemma 6, $|\mathcal{N}_k| \leq \log(K/\zeta)/\gamma_k + 1$. By Corollary 12, under $E_4$, $T \leq 2\text{OPT}_{\delta/2}^{\gamma_{k+1}}$ when PRECHECK is run for iteration $k$. Making these substitutions and reordering the terms gives the result. $\square$

**Lemma 15.** *For iterations $-1 \leq k < K - 1$, under $E_4$,* PRECHECK *runs for at most* $10\text{OPT}_{\delta/2}^{\gamma_{k+1}}\left\lceil 32.1\log\left(\frac{2K}{\zeta}\right)\right\rceil(\log(K/\zeta)/\gamma_k + 1)$ *time.*

*Proof.* By Corollary 12, under $E_4$, $T \leq 2\text{OPT}_{\delta/2}^{\gamma_{k+1}}$ when PRECHECK is run for iteration $k$. Phase I of PRECHECK is aborted when the total runtime reaches $1.9Tb' \leq 3.8\text{OPT}_{\delta/2}^{\gamma_{k+1}}b'$. Phase II of PRECHECK is aborted when the total Phase II runtime exceeds $2.99Tb' \leq 5.98\text{OPT}_{\delta/2}^{\gamma_{k+1}}b'$. This abort only happens after the last run, which takes at most $\tau'$ time, where $\tau'$ is measured in Phase I of

PRECHECK. Because of the way $\tau'$ is calculated by Phase I, at least $\lfloor 0.2b' \rfloor$ instances were running up until $\tau'$ time, which took $\lfloor 0.2b' \rfloor \tau' \leq 1.9Tb'$ time. For any valid setting of $\zeta$, $\lfloor 0.2b' \rfloor \geq 0.19b'$, so $\tau' \leq 10T \leq 20\text{OPT}_{\delta/2}^{\gamma_{k+1}} \leq 0.21\text{OPT}_{\delta/2}^{\gamma_{k+1}} b'$, so the work of PRECHECK for each configuration is upper bounded by $(3.8 + 5.98 + 0.21)\text{OPT}_{\delta/2}^{\gamma_{k+1}} b' < 10\text{OPT}_{\delta/2}^{\gamma_{k+1}} b'$. Multiplying this by the number of configurations $|\mathcal{N}_k|$ PRECHECK evaluates, and using Lemma 6, the total work of PRECHECK is bounded by $10\text{OPT}_{\delta/2}^{\gamma_{k+1}} b'(\log(K/\zeta)/\gamma_k + 1)$. $\qquad\square$

*Proof of Theorem 1.* Suppose $E_4$ and $E_5$ hold. By a union bound, taking also into account the lower bounds on the probabilities of these events (given by Lemma 10 and Lemma 14), we have $\Pr(E_4 \cap E_5) \geq 1 - 12\zeta$. By Corollary 11, under these events, the configuration returned by IMPATIENTCAPSANDRUNS is $(\varepsilon, \delta, \gamma)$-optimal.

Next we consider the runtime of IMPATIENTCAPSANDRUNS. For iteration $k = K - 1$, $E_1$, $E_2$, and $E_4$, by Corollary 13 and Lemma 6, the runtime of CAPSANDRUNS is $\tilde{\mathcal{O}}\left(b\text{OPT}_{\delta/2}^{\gamma_k}/\gamma_{K-1}\right)$. For iterations $0 \leq k < K-1$, by Corollary 13, the runtime of CAPSANDRUNS in iteration $k$ is upper bounded as $\tilde{\mathcal{O}}\left(b|\overline{\mathcal{N}}_k|\text{OPT}_{\delta/2}^{\gamma_{K-1}}\right)$. Using the bound $|\overline{\mathcal{N}}_k| = \tilde{\mathcal{O}}\left(F(38\text{OPT}_{\delta/2}^{\gamma_{k+1}})/\gamma_k\right)$ given by Lemma 14 the work by CAPSANDRUNS in iteration $k$ is bounded by $\tilde{\mathcal{O}}\left(bF(38\text{OPT}_{\delta/2}^{\gamma_{k+1}})\text{OPT}_{\delta/2}^{\gamma_k}/\gamma_k\right)$.

For the final CAPSANDRUNS call, the total work performed by CAPSANDRUNS would only increase if we didn't do any work on any configurations before, in other words, if we restarted CAPSANDRUNS with the input configurations $\overline{\mathcal{N}}$. By this idea we can upper bound the total work of the final CAPSANDRUNS call using [36, Theorem 1], which states that under $E_1$, the total work of a restarted CAPSANDRUNS with input configurations $\overline{\mathcal{N}}$ is at most $\tilde{\mathcal{O}}\left(|\overline{\mathcal{N}}|\frac{1}{\varepsilon^2\delta}\min_{i \in \overline{\mathcal{N}}} R^{\frac{\delta}{2}}(i)\right)$, which is a simplified form of the problem-dependent bound (1) in [36]. By Lemma 10, $\min_{i \in \overline{\mathcal{N}}} R^{\frac{\delta}{2}}(i) \leq \text{OPT}_{\delta/2}^{\gamma}$, and by Lemma 14, $|\overline{\mathcal{N}}| = \tilde{\mathcal{O}}\left(F(38\text{OPT}_{\delta/2}^{\gamma})/\gamma\right)$. Plugging these in the bound we get that the final CAPSANDRUNS takes $\tilde{\mathcal{O}}\left(\text{OPT}_{\delta/2}^{\gamma}F(38\text{OPT}_{\delta/2}^{\gamma})\frac{1}{\varepsilon^2\delta\gamma}\right)$ time.

Now we turn our attention to bounding the work done in PRECHECK. By Lemma 15, under $E_1$, for all the iterations, and including the final PRECHECK call, the total work is $\tilde{\mathcal{O}}\left(\sum_{k=0}^{K-1} \text{OPT}_{\delta/2}^{\gamma_k}/\gamma_k\right)$.

Adding all this work up, noting that $b = \tilde{\mathcal{O}}(1/\delta)$, we get that under $E_4$ and $E_5$, the total work performed by IMPATIENTCAPSANDRUNS is

$$\tilde{\mathcal{O}}\left(\frac{1}{\varepsilon^2\delta\gamma}\text{OPT}_{\delta/2}^{\gamma}F(38\text{OPT}_{\delta/2}^{\gamma}) + \sum_{k=0}^{K-2}\frac{1}{\gamma_k}\text{OPT}_{\delta/2}^{\gamma_k}\left(1 + F(38\text{OPT}_{\delta/2}^{\gamma_{k+1}})/\delta\right) + \frac{1}{\delta\gamma_{K-1}}\text{OPT}_{\delta/2}^{\gamma_{K-1}}\right).$$

$\qquad\square$

## B  Comparison of methods with theoretical guarantees

| Configuration algorithm | Anytime | No assumption on the maximum runtime | Capped OPT | Problem-dependent bound | Needle-in-haystack speedup |
|---|---|---|---|---|---|
| SP | ✓ | ✗ | ✗ | ✗ | ✗ |
| SPC | ✓ | ✓ | ✗ | ✗ | ✗ |
| LAB | ✗ | ✓ | ✗ | ✗ | ✗ |
| CAR | ✗ | ✓ | ✓ | ✓ | ✗ |
| ICAR | ✗ | ✓ | ✓ | ✓ | ✓ (up to $\gamma$-factor) |

Table 2: Comparison of algorithm-configuration methods with theoretical guarantees.

To select a near-optimal configuration with high probability from a finite set of $n$ configurations, the runtime of SP [24] can be bounded by $\tilde{\mathcal{O}}(n\frac{\text{OPT}}{\varepsilon^2\delta}\log\log\bar{\kappa})$, which includes a doubly logarithmic dependence on an upper bound on the maximum runtime $\bar{\kappa}$, which is a parameter of the algorithm and is assumed to be finite. While not anytime, LAB [35] achieves a runtime bound of $\tilde{\mathcal{O}}(n\frac{\text{OPT}}{\varepsilon^2\delta})$ with a

different method, independently of the maximum runtime (which can be infinite, and the algorithm does not need to have access to an upper bound on it). SPC [25] has a similar worst-case bound, and it improves upon SP by employing confidence bounds on the measurements, similarly to LAB. [24] also showed a lower bound of $\Omega(n\frac{\text{OPT}}{\varepsilon^2\delta})$ in the worst case, matching the upper bounds up to logarithmic terms. CAR [36] improves on LAB in two respects: (i) it finds a near-optimal configuration relative to $\text{OPT}_{\frac{\delta}{2}}$, the smallest $\delta/2$-capped runtime over the $n$ configurations, instead of the uncapped OPT; (ii) it enjoys a problem-dependent runtime bound that is much more favourable than the worst-case bound if the variances of the runtime distributions are better than in the worst case. The latter bound also scales with $\text{OPT}_{\frac{\delta}{2}}$ instead of OPT (the capping can be chosen to be arbitrarily close to $\delta$ but it must be smaller, as discussed in Section 2). This difference is significant, as there can be an arbitrarily large gap between $\text{OPT}_{\frac{\delta}{2}}$ and OPT.[9] Denoting the relative variance and relative range of the capped runtimes for configuration $i$ by $\hat{\sigma}^2(i)$ and $r(i)$, respectively (with respect to the expected capped runtime), and the relative gap between configuration $i$ and the optimal configuration by $\Delta_i$,[10] the runtime of CAR is bounded by

$$\tilde{O}\left(\text{OPT}_{\frac{\delta}{2}}\left[\frac{n}{\delta} + \sum_{i\in\mathcal{N}}\max\left\{\frac{\max\left\{\hat{\sigma}^2(i),\hat{\sigma}^2(i_*)\right\}}{\max\{\varepsilon^2,\Delta_i^2\}}, \frac{\max\left\{r(i),r(i_*)\right\}}{\max\{\varepsilon,\Delta_i\}}\right\}\right]\right).$$

This bound is always as good as the upper bounds above and matches the worst-case lower bound (up to logarithmic factors). Similarly to CAR, the guarantees for ICAR also involve the capped optimal runtime. Furthermore, a similar problem-dependent bound can be calculated straightforwardly for ICAR, which was omitted to simplify the presentation.

The runtimes of all the methods presented scale with $n$, the number of configurations. To convert to a guarantee of finding a near-optimal configuration from the top $\gamma$ fraction of an infinite pool, these methods select $n = \tilde{\mathcal{O}}(\gamma^{-1})$ configurations, and thus the runtimes scale with $\gamma^{-1}$. This work focuses on reducing this factor by "impatiently" eliminating configurations quickly. Table 2 summarizes the features of the different methods.

## C  Runtime Bound for Exponential Distributions

To better understand the runtime bound in Theorem 1, consider a scenario where the runtime of each configuration follows an exponential distribution. Such scenarios are realistic and motivated by practical applications [17]: roughly speaking, many solvers for NP-hard problems (e.g., SAT) proceed by initializing with a random seed and, if they fail, try again with another random seed. To better understand the runtime bound in Theorem 1, consider a scenario where the runtime of each configuration follows an exponential distribution. Such scenarios are realistic and motivated by practical applications [15, 17, 16, 26]: roughly speaking, many solvers for NP-hard problems (e.g., SAT) proceed by initializing with a random seed and, if they fail, try again with another random seed. To make the example concrete, suppose that the mean runtime for each configuration is distributed uniformly between $A$ and $A + B$, denoted by $U(A, A + B)$, for some $A, B > 0$. Here $A$ can be thought of as a small average runtime associated with the cost of starting the run of any configuration on any problem instance, and $B$ as the maximum "true" mean runtime of the configurations.

We can simplify the runtime bound of Theorem 1 for this setting. The best $\gamma_k$ fraction of the configurations have mean $A + B\gamma_k$ so $\text{OPT}_{\delta/2}^{\gamma_k} \leq A + B\gamma_k$. Furthermore, for a configuration $i$ with mean $\frac{1}{\lambda}$, $R_\tau(i) = \frac{1}{\lambda}\left(1 - e^{-\lambda\tau}\right)$ for any runtime cap $\tau$. Substituting $\tau = t_\delta(i)$, noting that the probability of running over the cap is $\delta$ so $e^{-\lambda\tau} = \delta$, we have $R^\delta(i) = \frac{1}{\lambda}(1 - \delta)$. Similarly, $R^{0.35}(i) = 0.65\frac{1}{\lambda}$. Then $F(38\text{OPT}_{\delta/2}^{\gamma_k}) - 4\zeta/K = \text{Pr}_{i\sim\Pi}(R^{0.35}(i) \leq 38\text{OPT}_{\delta/2}^{\gamma_k}) \leq \text{Pr}_{\frac{1}{\lambda}\sim U(A,A+B)}(0.65\frac{1}{\lambda} \leq 38(A + B\gamma_k)) \leq \text{Pr}_{\frac{1}{\lambda}\sim U(A,A+B)}(\frac{1}{\lambda} \leq 58.5(A + B\gamma_k)) \leq \frac{58.5(A+B\gamma_k)-A}{B} = \mathcal{O}(\gamma_k + \frac{A}{B})$. This bounds $F(38\text{OPT}_{\delta/2}^{\gamma_k}) - 4\zeta/K$. The extra $4\zeta/K$ is insignificant, as the failure probability $\zeta$ can simply be chosen to be $\mathcal{O}(\varepsilon^2\delta)$, resulting in only additional logarithmic factors in the runtime; thus, any term multiplied by $\zeta$ in the runtime bound is of such low order and disappears in the $\tilde{\mathcal{O}}(\cdot)$ notation. Substituting the bound on $F$ and $\text{OPT}_{\delta/2}^{\gamma_k}$, assuming a choice of $\zeta$ as above, the runtime bound of

Theorem 1 becomes

$$\tilde{\mathcal{O}}\left(\frac{A+B\gamma}{\varepsilon^2\delta\gamma}\cdot\left(\gamma+\frac{A}{B}\right)+\sum_{k=0}^{K-2}\frac{A+B\gamma_k}{\gamma_k}\left(1+\frac{\left(\gamma_k+\frac{A}{B}\right)}{\delta}\right)+\frac{A+B\gamma_{K-1}}{\delta\gamma_{K-1}}\right)$$
$$=\tilde{\mathcal{O}}\left(\frac{1}{\varepsilon^2\delta}\left(\frac{A^2}{B\gamma}+A+B\gamma\right)\gamma+\frac{1}{\delta}\left(\frac{A}{\gamma_{K-1}}+B\right)+\frac{A}{\gamma}\right),$$

where the $K$ term disappears in the $\tilde{\mathcal{O}}(\cdot)$ notation as $K\le\log_2(\frac{1}{\gamma})$ and we also used that $\gamma_k=\gamma 2^k$ for $k\in[0,K-1]$. Contrasting this with the typical runtime bound $\tilde{\mathcal{O}}\left(\frac{1}{\varepsilon^2\delta\gamma}\mathrm{OPT}_{\delta/2}^\gamma\right)=\tilde{\mathcal{O}}\left(\left(\frac{A}{\gamma}+B\right)\frac{1}{\varepsilon^2\delta}\right)$ of previous works, we can see that the main term (the one multiplied by $\frac{1}{\varepsilon^2\delta}$) is reduced by a factor of $\max\{\gamma,\frac{A}{B}\}$. The rest of the terms have no dependence on $\varepsilon$ and are typically always much smaller (under reasonable parameter values, essentially if $B/A$ is not too large compared to $1/\varepsilon^2$) than the typical runtime bound for other works: , e.g., $\frac{A}{\delta\gamma_{K-1}}$ is smaller than the first term provided $\varepsilon^2 B/A$ is small compared to $2^{K-1}$ (which holds if $K$ is large enough or $\varepsilon$ is small enough); $\frac{B}{\delta}$ does not depend on the number of configurations evaluated neither does it scale with $\varepsilon$. Note that the last term, $\frac{A}{\gamma}$, is associated with having to evaluate about $\frac{1}{\gamma}$ number of configurations, and this term could not scale better than with the minimum runtime $A$ (this term again is small unless $B/A$ is huge, on the order of $1/(\varepsilon^2\delta)$).

## D  Details of Experiments

We followed the experimental setup of Weisz et al. [36]. Runs were pre-computed and then queried from a simulation environment in which they can be stopped and resumed. In a scenario where this is not possible (e.g., due to memory constraints when performing real runs) the experiments can still be implemented by restarting runs from scratch with doubling cap times, resulting in at most a factor of 2 slowdown.

**Parameter values**  Experiments on all datasets were done with $(\varepsilon,\delta)=(0.05,0.1)$ and varying $\gamma\in\{0.01,0.02,0.05\}$. For each configurator, $\zeta$ was set so that the total failure probability is $0.05$. The hyperparameter $K$ was set such that $0.25<\gamma 2^{K-1}\le 0.5$. This is a somewhat arbitrary choice, but was made so that values of $\gamma_k$ were neither too big to be trivial, nor too small to be computationally prohibitive.

**EPM Setup**  We used the provided generators for Regions200 and RCW to produce as many new random instances as needed, which were pre-processed using the feature extractors provided with the EPM.[11] Runtime-related features (e.g., CPU time required for feature computation) were dropped since they are machine-dependent. We then used the provided configurations and runtime data[12] to train the EPM model, using the parameters suggested in [13]. Finally, the trained EPM was used as a surrogate model to provide runtimes for new configuration-instance pairs. New configurations were sampled by uniformly choosing a value for each parameter of CPLEX from the appropriate range. A new instance was then generated and the pair was given to the EPM. Note that ICAR examined more configurations than CAR did. For consistency, sampling was done so that the configurations given to CAR and ICAR were streamed in the same order. Consequently, the configurations seen by CAR were a subset of those seen by ICAR. To aid future experimentation, we pre-generated and stored the runtime data (see details below), which we make available[13] in addition to our EPM pipeline code.[14]

**Description of the datasets**

Figure 2: Distribution of $\delta$-capped mean runtime of the sampled configurations, with $\delta = 0.1$. For Minisat/CNFuzzDD, many configurations are close to the optimal one, whereas for CPLEX/Regions200 and CPLEX/RCW, many configurations are significantly worse than the optimal one. Consequently, PRECHECK is able to discard more configurations in the latter two scenarios.

- Minisat/CNFuzzDD is a SAT scenario based on the minisat solver, with 6 parameters, applied to the CNFuzzDD[15] instances. The benchmark dataset[16] we used is the same as in [35, 36, 25], including 972 configurations and 20118 instances. To simulate the process of sampling random configurations, we randomly selected an unused configuration from the pool upon request from CAR/ICAR. Again, the selection was made such that they were given to CAR and ICAR in the same order.

- CPLEX/Regions200 is a MIP scenario using CPLEX, an interger programming solver, applied to a combinatorial auction winner determination problem. For CPLEX, we used the same configuration space as in [13]: There are 74 parameters, where categorial and small-domain integeral/continuous parameters were sampled uniformly, and large-domain integral/continuous parameters were sampled log-uniformly. The benchmark dataset is generated with the EPM described above, with 10000 configurations and 50000 instances. It takes around 13 CPU days on a single thread to generate the datasets on our machines (Intel Core i7-7700K).

- CPLEX/RCW is a MIP scenario using the CPLEX solver applied to Red-cockaded Woodpecker conservation problems. The configuration space is the same as CPLEX/Regions200. The benchmark dataset is generated with the EPM described above, with 10000 configurations and 35640 instances. It takes around 20 CPU days on a single thread to generate the datasets on our machines.

### D.1  IMPATIENTCAPSANDRUNS **with Varying Parameters**

We also compared the performance of ICAR and CAR++ with varying values of $\varepsilon$ and $\delta$ (with fixed $\gamma = 0.02$ and failure probability $0.05$). The speedup (computed as the ratio of the runtimes) achieved by ICAR over CAR++ is reported in Table 3. As we can see, for the CPLEX datasets, the speedup was fairly stable across a range of $\varepsilon$ and $\delta$ that a user might be likely to care about. Table 4 shows the ratio of the $\delta$-capped mean runtime of the returned configurations. On the other hand, for Minisat/CNFuzzDD, CAR++ was up to 20% faster than ICAR. As we can see, ICAR and CAR++ returned configurations with very similar quality, but ICAR sometimes returned slightly better ones. To understand the different behavior in the different datasets, a histogram of the runtime distributions over the configurations is plotted in Figure 2, showing that in case of Minisat/CNFuzzDD, there are much more near-optimal configurations than for CPLEX/Regions200 and CPLEX/RCW, making the early discard procedure much less effective.

### D.2  **Synthetic Experiments**

To better understand how well ICAR can exploit a needle-in-a-haystack scenario, we examined its performance on synthetic data. In this way we could choose each configuration's true mean, and thus control their distribution. The runtimes of each configuration were sampled from an exponential distribution, with the means being uniformly chosen from the interval $[\text{OPT}, \ c \cdot \text{OPT}]$. We tend to

|  | Minisat/CNFuzzDD | | | | CPLEX/Regions200 | | | | CPLEX/RCW | | | |
|---|---|---|---|---|---|---|---|---|---|---|---|---|
| $\delta$ | 0.025 | 0.05 | 0.075 | 0.1 | 0.025 | 0.05 | 0.075 | 0.1 | 0.025 | 0.05 | 0.075 | 0.1 |
| $\varepsilon = 0.025$ | 0.80 | 0.83 | 0.71 | 0.95 | 2.76 | 2.44 | 2.15 | 2.03 | 2.27 | 1.99 | 1.83 | 1.63 |
| $\varepsilon = 0.05$ | 0.83 | 0.84 | 0.78 | 0.92 | 2.90 | 2.54 | 2.24 | 2.06 | 2.54 | 2.21 | 1.96 | 1.80 |
| $\varepsilon = 0.075$ | 0.84 | 0.86 | 0.82 | 0.92 | 2.94 | 2.58 | 2.28 | 2.09 | 2.66 | 2.30 | 2.03 | 1.85 |
| $\varepsilon = 0.1$ | 0.85 | 0.87 | 0.85 | 0.93 | 2.97 | 2.60 | 2.29 | 2.11 | 2.73 | 2.36 | 2.08 | 1.88 |

Table 3: Speedup achieved by ICAR over CAR++ for various values of $\varepsilon$ and $\delta$. The runtimes of ICAR and CAR++ are averaged over five runs. Values greater than one indicate ICAR is faster.

|  | Minisat/CNFuzzDD | | | | CPLEX/Regions200 | | | | CPLEX/RCW | | | |
|---|---|---|---|---|---|---|---|---|---|---|---|---|
| $\delta$ | 0.025 | 0.05 | 0.075 | 0.1 | 0.025 | 0.05 | 0.075 | 0.1 | 0.025 | 0.05 | 0.075 | 0.1 |
| $\varepsilon = 0.025$ | 1.00 | 1.00 | 1.00 | 1.00 | 0.93 | 0.93 | 0.93 | 0.93 | 1.00 | 0.99 | 0.98 | 0.98 |
| $\varepsilon = 0.05$ | 1.00 | 1.00 | 1.00 | 1.00 | 0.93 | 0.93 | 0.93 | 0.93 | 1.00 | 0.99 | 0.98 | 0.98 |
| $\varepsilon = 0.075$ | 1.00 | 1.00 | 1.00 | 1.00 | 0.93 | 0.93 | 0.93 | 0.93 | 1.00 | 0.99 | 0.98 | 0.98 |
| $\varepsilon = 0.1$ | 1.00 | 1.00 | 1.00 | 1.00 | 0.93 | 0.93 | 0.93 | 0.93 | 1.00 | 0.99 | 0.98 | 0.98 |

Table 4: Ratio of $\delta$-capped mean runtime of the configurations returned by ICAR over CAR++ for various values of $\varepsilon$ and $\delta$. The $\delta$-capped mean runtime of the returned configurations are averaged over five runs. Values smaller than one indicate ICAR returned better solutions.

|  | Total CPU Time (days) | | | | Number of Configurations Before/After Precheck | | | |
|---|---|---|---|---|---|---|---|---|
|  | $c = 2$ | $c = 5$ | $c = 10$ | $c = 25$ | $c = 2$ | $c = 5$ | $c = 10$ | $c = 25$ |
| ICAR | 505 (23) | 187 (18) | 113 (17) | 92 (27) | 351 / 349 | 351 / 114 | 351/ 54 | 351/ 27 |
| CAR++ | 344 (24) | 214 (12) | 193 (20) | 195 (31) | 245 | 245 | 245 | 245 |
| CAR | 447 (21) | 384 (15) | 380 (35) | 406 (62) | 245 | 245 | 245 | 245 |

Table 5: Total CPU time in days to find a $(0.05, 0.1, 0.02)$-optimal configuration and the number of configurations before and after precheck in the synthetic experiments. For CAR and CAR++, the number of configurations sampled is reported. CAR++ is the improved version CAR arising from more careful analysis. Error terms are standard deviations over five runs.

think that real algorithm runtimes do look somewhat exponential, and there is justification for this, at least in certain cases [15, 17, 16, 26].

We ran the simulation for $c \in \{2, 5, 10, 25\}$. The larger the value of $c$, the more configurations will tend to be far from the best one, creating more and more of a "needle-in-a-haystack" scenario. We used $(\varepsilon, \delta, \gamma) = (0.05, 0.1, 0.02)$ and failure probability of $0.05$, as before. Table 5 shows the total CPU time to find a $(0.05, 0.1, 0.02)$-optimal configuration for the range of $c$ values. The degree to which ICAR outperformed CAR and CAR++ increases as $c$ increases, as expected. We can see that PRECHECK was able to reject a greater proportion of configurations when many tended to be far from optimal (large $c$).

Figure 3 shows the CPU time spent on each configuration, sorted by the $\delta$-capped mean runtime. When $c = 2$, ICAR rejected very few configurations in PRECHECK, but as $c$ increases we can see a greater proportion of configurations were being run for minimal time compared to CAR++. Furthermore, the runtime of CAR and CAR++ became dominated by the runs on the bad configurations, as those configurations contribute a large amount to the area under the curves.

Figure 3: Synthetic experiments: As the proportion of configurations that are far from the optimal gets larger (i.e., as $c$ gets larger), the CPU runtime of CAR was more dominated by the work spent on bad configurations, while ICAR was able to drop more bad configurations with its PRECHECK mechanism. Note the log scale on the $y$-axis.

# E   High-Probability Tail Bounds

For convenience, we summarize here the main concentration inequalities used in the paper. For proofs, see, e.g., [12] and [5].

**Bernstein inequality**   Let $X_1, \ldots, X_n$ be independent zero-mean random variables with range $R$ (i.e., $|X_i| \leq R$ almost surely, for all $i$). Then, for any $t > 0$,

$$\Pr\left(\sum_{i=1}^{n} X_i \geq t\right) \leq \exp\left(-\frac{\frac{1}{2}t^2}{\sum_{i=1}^{n} E(X_i^2) + \frac{1}{3}Rt}\right).$$

**Empirical Bernstein bound**   Let $X_1, \ldots, X_n$ be independent and identically distributed random variables with range $R$ and mean $\mu$. Let the empirical mean be $\bar{X}$ and the empirical variance be $\bar{\sigma}^2 = \frac{1}{n}\sum_{i=1}^{n}(X_i - \bar{X})^2$. Applying Bernstein's inequality to the sum and the sum of the squares of these random variables, we get the empirical Bernstein bound [4, 5], which states that with probability at least $1 - \zeta$,

$$|\bar{X} - \mu| \leq \sqrt{\frac{2\bar{\sigma}^2 \log(3/\zeta)}{n}} + \frac{3R\log(3/\zeta)}{n}.$$

**Chernoff bound**   Let $X$ be a set of $n$ independent and identically distributed Bernoulli random variables. Let their empirical average be $\bar{X} = \frac{1}{n}\sum_{i]1}^{n} X_i$ and let $E(X_i) = \mu$. Then,

- $\Pr\left(\bar{X} \geq (1+c)\mu\right) \leq \exp\left(-\frac{c^2}{2+c}n\mu\right)$ for any $c > 0$, and
- $\Pr\left(\bar{X} \leq (1-c)\mu\right) \leq \exp\left(-\frac{c^2}{2}n\mu\right)$ for any $0 < c < 1$.