[Reviews · NeurIPS 2020]

Review 1

Summary and Contributions: This paper considers the problem of algorithm configuration: finding a near-optimal set of parameters for an algorithm that minimizes the average runtime over inputs sampled from a given distribution. The paper presents a new algorithm configuration method called IMPATIENTCAPSANDRUNS (ICAR), which aims to address the performance gap between heuristic methods for algorithm configuration that perform well in practice but have no performance guarantees and theoretical methods that have performance guarantees but run slower. Specifically, the paper improves on the existing CAR algorithm by introducing an ‘impatient’ PRECHECK mechanics, often used in heuristic approaches, that discards less promising configurations early on to reduce overall runtime. The paper considers instances from two domains: Mixed Integer Programming (MIP) and Satisfiability (SAT). Post rebuttal: ========== I read the authors’ response. Thank you for the additional results and explanations!

Strengths: ICAR integrates an approach common in heuristic algorithm configuration methods into a theoretically oriented approach. The main strength of the paper is the theoretical result, which is intuitive, well-motived, and relevant to the community. The idea of introducing the “precheck” mechanism is simple, but the authors are able to provide theoretical guarantees on their proposed procedure. Moreover, according to Table 1 and Figure 1, the practical performance of the proposed procedure comparing with baselines seems promising. The research makes strides towards closing the gap between theoretical and empirical algorithm configuration, which is important in practice.

Weaknesses: The paper is mainly theoretical; its main weakness is the relatively small empirical evaluation. As the authors claim to close the gap relative to empirical heuristics, one would expect more evaluation. It would be preferable to include at least one heuristic method to see how close this method gets to closing the theoretical/practical gap. Additionally, the explanation of the algorithms on page 5 is relatively dense and might not be accessible outside domain experts. Since the paper is of practical interest, it would be preferable to move some algorithmic details to the appendix and present a more high-level version of the algorithm in the main paper.

Correctness: The theoretical claims seem sound. Regarding empirical evaluation, it would be useful to consider more than three types of instances, and to compare against some additional heuristic baselines. Additionally, it would be useful to see how well the EPM models perform and to comment on how the accuracy of the model might impact the experiments. Also, for SAT, are the instances sat or unsat?

Clarity: The paper is well written. Page 5 is a little dense, and it might be preferable to move some algorithms to the appendix and use the space for extended empirical evaluation. Line 697 should probably use /citet and not /cite. There seems to be a typo on line 662 - it says “««< HEAD”.

Relation to Prior Work: The relationship to previous work is clearly outlined. I would suggest a table in the appendix that lists the theoretical performance guarantees of previous work and this paper, which would make it easier to compare the methods.

Reproducibility: Yes

Additional Feedback: It would be interesting if the authors could comment on how their results might be applicable to machine learning and deep learning, which is also very sensitive to hyperparameters. The authors commit to releasing precomputed run-times, will the code for experiments also be released?


Review 2

Summary and Contributions: This paper proposes a new method for finding good algorithm configuration with small runtimes. The authors try to address the problem of the existing methods: theory-based methods are conservative and discard less promising configurations until they demonstrate its ineffectiveness in theory, which makes it perform much worse than some heuristic methods. The proposed method leverages the high-level idea of the CAR algorithm, and introduces a “quick discard” mechanism to allows poorly performing configurations to be discarded quickly. The authors verify the superior performance of the proposed methods than baselines in both theory and experiment. ############ I would like to keep my score after reading authors' response.

Strengths: This paper proposes a new method for algorithm configuration. This paper establishes the theoretical guarantee of the proposed method, which demonstrates its improved performance compared to CAR.

Weaknesses: The comparison to the theoretical guarantees of the existing methods is not clearly presented. In the experiment, the authors should also compare the runtime of the proposed method with heuristic methods, rather than only focusing on the methods with theoretical guarantees (e.g., CAR, CAR++).

Correctness: yes

Clarity: yes

Relation to Prior Work: yes

Reproducibility: Yes

Additional Feedback: The comparison in terms of theoretical guarantees should be delivered in a clearer way. It would be better to include a table that states the complexity of all related methods so that the reader can easily see the difference and advantange of the proposed methods. In experiments, it would be better to include the comparison with some heuristic methods and compare their runtimes and optimality of the obtained solutions. In line 134-136, the author states that one can simply select a pool of log(zeta)/log(1-gamma) configures to ensure with prob. at least 1-zeta it contains a configuration that belongs to the top gamma-fraction of all the ones. Could you please provide more details to justify this argument? In the introduction, the authors would like to summarize the main contribution, and clearly identify the differences and improvements compared to existing methods. Additionally, briefly introduce some technical details in the introduction section can also help the reader understand why ICAR can still enjoy favorable optimality after adding the precheck mechanism.


Review 3

Summary and Contributions: This paper studies the problem of algorithm configuration. The authors propose a method named ImpatientCapsAndRuns, which quickly discards less promising configurations, significantly speeding up the search procedure with theoretical guarantees and still achieving optimal runtime up to logarithmic factors. The experimental results also validate the effectiveness of the proposed method.

Strengths: The contribution is significant and novel: This paper provides an improved method over the recent work CapsAndRuns [35]. The improvement is inspired by the idea of the heuristic and practical procedures discarding parameters “impatiently” based on very few observations. The authors provide a theoretical guanrantee for the proposed method, which can show the advantages comparing to existing methods. The method in this paper applies an initial “precheck” mechanism that allows poorly performing configurations to be discarded quickly. The experiment results show that it can outperform existing methods significantly.

Weaknesses: Below Theorem 1, the authors have already presented some theoretical comparisons between ImpatientCapsAndRuns and CapsAndRuns [35]. However, there is still a little difficulty to directly observe the improvement of the result in Theorem 1 of this paper over the result in Theorem 1 of [35] due to some different notations and definitions. Can the authors provide further intuitive explanations about how to compare the two results?

Correctness: The theoretical claims look correct. The detailed proofs are not carefully checked. The empirical methodology is correct.

Clarity: This paper is well organized and has provided sufficient details in both theoretical part and experimental part.

Relation to Prior Work: The authors have clearly compared their results with previous works.

Reproducibility: Yes

Additional Feedback:


Review 4

Summary and Contributions: The paper proposes a new method for finding algorithm configuration with theoretical performance guarantee (Theorem 1). The proposed algorithm ICAR can be seen as a valiant of CapsAndRuns. The main contribution of the paper is summarized in Theorem 1 which reveals time complexity of ICAR. Numerical experiments supports efficiency of the proposed method.

Strengths: The most notable contribution of this paper is theoretical guarantee for total running time (Theorem 1) for ICAR. The equation (1) explicitly presents dependency of epsilon, delta, gamma to the total time bound. The ICAR is a practically imprementable and its effectiveness is supported by numerical experiments on MIP and SAT problems. I think it is a good contribution for reducing the search time of general hard problems with huge parameter spaces.

Weaknesses: In the results of numerical experiments (Figure 1), ICAR mostly shows the least complexity compared with the baselines but for some cases (spikes) , the time complexity is larger than that of CAR++. ========== After Authors' Rebuttal ========== I read the authors' response and fully understand their explanation.

Correctness: The discussion of this paper seems quite reliable.

Clarity: The manuscript is well written.

Relation to Prior Work: Related works are well discussed.

Reproducibility: Yes

Additional Feedback:

[Author Response · NeurIPS 2020]

We would like to thank the reviewer for their insightful comments. Below we address the main concerns:

**To Reviewers 1 & 2:** Comparison with heuristic methods:

Making a meaningful comparison with heuristic methods at this stage
is a hard: While our algorithm is designed to stop when a near-optimal
configuration is found provably, the method can continuously provide a
candidate configuration, and it is not clear at which time point one should
make the comparison (and what $\varepsilon$ and $\delta$ parameters should be selected).
Nevertheless, Figure 1 shows (for Minisat with $\gamma = 0.05$) that the best
configuration is found much earlier than the time needed to prove that it
is indeed (near-)optimal (19 vs. 101 CPU-days).

**To Reviewer 1:**

Figure 1: Actual expected runtime and the corresponding high probability bound maintained by ICAR for the top-ranked configuration (Minisat, $\gamma = 0.05$).

• More detailed high-level description of the algorithm: We will try our
best to fit a more detailed high-level description of the algorithm in the
main text.

• EPM: The accuracy of the EPM model has been analyzed thoroughly
in [12]. Any controllable (additive or multiplicative) error of the sim-
ulator can easily be incorporated in the analysis. Of course, when the
EPM model exhibits poor performance, the practical performance can
degrade, but to guarantee the correctness of the algorithm, we only need a good accuracy of the EPM approximation
near the optimum, although the runtime can be increased if additional variance of the runtime distribution is
introduced by the simulator.

• sat vs unsat: The Minisat dataset contains both satisfiable and unsatisfiable instances.

**To Reviewer 2:**

• Presentation: For ease of reference, we will improve the presentation of the theoretical bounds of previous work as
recommended. Nevertheless, the order of these bounds ($\frac{\mathrm{OPT}_{\delta/2}^{\gamma}}{\varepsilon^2 \delta \gamma}$) is presented in discussions (i) and (iii) following
Theorem 1, and also in Appendix B. We will also try our best to accommodate the suggested changes to the
introduction.

• Pool of size $n = \log(\zeta)/\log(1-\gamma)$: every randomly selected configuration is in the top $\gamma$ fraction of all the
configurations with probability $\gamma$. Hence, the probability that no configuration out of $n$ is top-$\gamma$ is $(1-\gamma)^n = \zeta$
(see also the proof of Lemma 4).

**To Reviewer 3:**

Theoretical comparison to CAR: The bound for CAR [35, eq. 1] is much more complex because it is a problem-
dependent bound which includes, e.g., the relative variance of the runtimes for each configuration and the gaps between
the expected runtime of the best configuration and the other configurations. This bound in the worst case (in the
minimax sense) simplifies to $\tilde{\mathcal{O}}\left(\frac{n\mathrm{OPT}_{\delta/2}}{\delta\varepsilon^2}\right)$ (where $n$ is the number of configurations), which is essentially the same
as the first (dominant) term of (1) in our paper without the factor $F$ (and replacing $\mathrm{OPT}_{\delta/2}$ with $\mathrm{OPT}_{\delta/2}^{\gamma}$, using the
number of instances $n_I = \log(K/\zeta)/\gamma$ ICAR uses). In fact, if one follows the more complicated derivation of [35], the
first term in (1) can be replaced with a similar problem-dependent summation as in [35, eq. 1], however, only over the
configurations not rejected by precheck (i.e., over at most $n_I F(38\mathrm{OPT}_{\delta/2}^{\gamma})$ configurations, and replacing $n$ in the first
term of [35, eq. 1] with the same quantity), while the rest of the terms remain the same (an added complication here is
that the gaps and relative variances would become random variables in our case, with a tricky dependence introduced by
the precheck mechanism). We omitted this analysis for simplicity and for the clarity of the presentation.

**To Reviewer 4:**

Spikes in Figure 1 (of the paper): Note that this figure shows the runtime of individual configurations, thus the fact that
ICAR runs some configurations longer than CAR++ is not indicative of their overall performance (nevertheless, as
shown in Table 1, the total runtime of CAR++ can be slightly smaller than that of ICAR when the runtime distribution
is simple, and there is no real need to run our carefully designed precheck method).

[Meta-Review · NeurIPS 2020]

The reviewers were overall positive about this paper. They felt that it gives a good theoretical contribution for an important problem. The only concern was that the experimental evaluation could be more substantial.